# Precise solid-phase synthesis of CoFe@FeO$_x$ nanoparticles for efficient polysulfide regulation in lithium/sodium-sulfur batteries

Yanping Chen [1,8], Yu Yao [2,8], Wantong Zhao[3], Lifeng Wang[2], Haitao Li[1], Jiangwei Zhang [4], Baojun Wang[3], Yi Jia [5], Riguang Zhang [3]✉, Yan Yu [2]✉ & Jian Liu [1,4,6,7]✉

Complex metal nanoparticles distributed uniformly on supports demonstrate distinctive physicochemical properties and thus attract a wide attention for applications. The commonly used wet chemistry methods display limitations to achieve the nanoparticle structure design and uniform dispersion simultaneously. Solid-phase synthesis serves as an interesting strategy which can achieve the fabrication of complex metal nanoparticles on supports. Herein, the solid-phase synthesis strategy is developed to precisely synthesize uniformly distributed CoFe@FeO$_x$ core@shell nanoparticles. Fe atoms are preferentially exsolved from CoFe alloy bulk to the surface and then be carburized into a Fe$_x$C shell under thermal syngas atmosphere, subsequently the formed Fe$_x$C shell is passivated by air, obtaining CoFe@FeO$_x$ with a CoFe alloy core and a FeO$_x$ shell. This strategy is universal for the synthesis of MFe@FeO$_x$ (M = Co, Ni, Mn). The CoFe@FeO$_x$ exhibits bifunctional effect on regulating polysulfides as the separator coating layer for Li-S and Na-S batteries. This method could be developed into solid-phase synthetic systems to construct well distributed complex metal nanoparticles.

Metal nanoparticles (NPs) with complex structure and uniform distribution demonstrate distinctive physicochemical properties and are widely applied in many fields. The development of their synthesis strategies benefit the exploration of more applications[1,2]. The commonly used synthesis approaches of metal NPs are the traditional wet chemistry methods, such as thermal decomposition, precipitation, hydrothermal/solvothermal, microemulsion, and sol-gel methods[1,3–5]. Metal NPs with controlled structure have been successfully synthesized by the thermal decomposition method, such as Co nanorods[6], hollow CoS NPs[7,8], In$_2$O$_3$ nanoflowers[9], MnFe$_2$O$_4$ nanocubes[10], FePt Nanocubes[11], and FeCo NPs[12]. The precipitation methods are applied to synthesis Pd-Pt nanodendrites[13], Co$_3$O$_4$ nanorods[14], and PdPt nanocages[15]. Meanwhile, several synthesis mechanisms are proposed to investigate the structure modulation of metal NPs, including focusing of size distribution[16], Kirkendall effect[7], galvanic replacement[15], cation exchange[17], and limited ligand protection theory[9].

[1]State Key Laboratory of Catalysis, Dalian Institute of Chemical Physics, Chinese Academy of Sciences, Dalian, Liaoning 116023, China. [2]Hefei National Research Center for Physical Sciences at the Microscale, Department of Materials Science and Engineering, National Synchrotron Radiation Laboratory, CAS Key Laboratory of Materials for Energy Conversion, University of Science and Technology of China, Hefei, Anhui 230026, China. [3]State Key Laboratory of Clean and Efficient Coal Utilization, College of Chemical Engineering and Technology, Taiyuan University of Technology, Taiyuan, Shanxi 030024, China. [4]Science Center of Energy Material and Chemistry, College of Chemistry and Chemical Engineering, Inner Mongolia University, Hohhot 010021, China. [5]Department of Applied Chemistry and Zhejiang Carbon Neutral Innovation Institute, Zhejiang University of Technology, Hangzhou 310032, China. [6]DICP-Surrey Joint Centre for Future Materials, Department of Chemical and Process Engineering, and Advanced Technology Institute, University of Surrey, Guildford, Surrey GU2 7XH, UK. [7]Center of Materials Science and Optoelectronics Engineering, University of Chinese Academy of Sciences, Beijing 100049, China. [8]These authors contributed equally: Yanping Chen, Yu Yao. ✉e-mail: zhangriguang@tyut.edu.cn; yanyumse@ustc.edu.cn; jian.liu@surrey.ac.uk

Although the wet chemical methods control the metal NPs structure well, yet their liquid-phase operation environment leads to the wastage of solvent or water. The construction of a complicated structure needs multistep operation, such as the core@shell structure is realized through coating a shell outside the pre-prepared core[3]. It is hard to both achieve the structure design and uniform dispersion of metal NPs simultaneously[6,11,18]. Metal NPs with complex structure can be synthesized via solid-phase synthesis with thermal gas treatment. The $Fe_3C$ and $Co_3C$ nanocrystalline confined in graphitic shells are synthesized through a chemical vapor deposition method under a mixture of $H_2$, $H_2O$, and $CH_4$ at 850 °C[19]. CoFe alloy NPs were formed by exsolution under a thermal reduction atmosphere[20]. The hollow Co-rich CoCu alloy structure was obtained through exposing the hollow CuCo alloy NPs to thermal syngas[21]. This thermal gas treatment process is operated under various gas at high temperatures without involving liquid-phase environment, displaying some attractive features, such as elimination of water or solvent wastage, facile operation, and simple synthesis parameters[22,23]. However, construction of metal NPs under thermal gas atmosphere sometimes is only considered as postprocessing means and has seldom been applied as an effective method to synthesize complex metal NPs. Therefore, constructing metal NPs with specific structure and exploring synthesis mechanism may promote the development of this thermal gas treatment method into a general synthesis strategy.

Herein, the $CoFe@FeO_x$ core@shell NPs are successfully prepared through the solid-phase synthesis strategy which is operated under thermal syngas ($H_2/CO$). The obtained $CoFe@FeO_x$ NPs are uniformly distributed on carbon matrix and demonstrate a CoFe alloy core and a $FeO_x$ shell. This strategy simultaneously realizes the synthesis of core@shell metal NPs and their uniform distribution on supports, serving as a potential synthetic strategy. The as-prepared core@shell NPs is expected to deliver great application prospect for energy storage field. $CoFe@FeO_x$ NPs could serve as the modifying layer of commercial separators for high-performance lithium-sulfur (Li-S) and sodium sulfur (Na-S) batteries, which is beneficial for solving the current challenge of commercial separators that can hardly suppress the polysulfide dissolution and shuttle issues[24,25]. The polar $FeO_x$ shell possesses strong adsorption ability to anchor polysulfides and the conductive CoFe core can facilitate the conversion process of polysulfides. As a result, the sulfur utilization and the cycling stability of Li-S and Na-S batteries are significantly enhanced due to the bifunctional effect of $CoFe@FeO_x$ NPs on regulating polysulfides. In particular, the Na-S battery with $CoFe@FeO_x$ modified separator delivers a high reversible capacity of 320 mAh g$^{-1}$ after 1200 cycles with nearly 100% Coulombic efficiency at 2 A g$^{-1}$.

## Results

### Synthesis and characterization of the $CoFe@FeO_x$

The experimental synthesis of the $CoFe@FeO_x$ starts from the pyrolysis of Co-Fe Prussian blue analogue (PBA). Co-Fe PBA is a kind of coordination polymer with $Fe^{3+}$ and $Co^{2+}$ bridged by the $CN^-$ groups. As shown in Fig. 1a, the CoFe@C with a CoFe alloy core and a carbon shell are obtained after the pyrolysis, which are well distributed on carbon matrix. The formed CoFe@C experience carbon shell falling off, Fe shell formation by exsolution, and $Fe_xC$ shell formation by carburization, producing the $CoFe@Fe_xC$ intermediate. The carburization occurs under syngas atmosphere at 240 °C. The final $CoFe@FeO_x$ are obtained through the passivation of $Fe_xC$ shell in air at room temperature. Scanning electron microscope (SEM) image of Co-Fe PBA displays uniformly distributed nanocubes morphology with ~200 nm of particle size (Fig. 1b). The diffraction peaks of Co-Fe PBA become disappearing and that of CoFe alloy emerge as the pyrolysis temperature increases from 25 to 700 °C (Fig. S1a). The high-resolution transmission electron microscopy (HRTEM) images demonstrate the formed CoFe@C nanoparticle consists of a CoFe alloy core and a

carbon shell (Fig. 1c), with ~60 nm of particle size (Fig. S1b). The core shows 2.01 Å of lattice distance, ascribing to the (110) planes of CoFe alloy (PDF#49-1568). The Raman spectra of CoFe@C exhibits two distinct peaks with $I_D/I_G$ of 1.00, indicating the shell presents a mixture of amorphous and graphitized carbon (Fig. S2). The formed $CoFe@FeO_x$ NPs show a core@shell structure, with ~5 nm of shell thickness (Fig. 1d). The Fast Fourier Transform (FFT) analysis displays three diffraction facets of the core and two diffraction facets of the shell, ascribing to CoFe alloy and $Fe_3O_4$ (PDF#79-0416) respectively.

The high-angle annular dark-field scanning transmission electron microscope (HAADF-STEM) and electron energy loss spectroscopy (EELS) elemental mapping images of $CoFe@FeO_x$ demonstrate that the core is mainly composed of Co and Fe elements and the shell mainly of Fe and O (Fig. 1e, S3, S4). It is noteworthy that, a passivating surface oxide formed at room temperature is often amorphous, beam-induced heating during TEM analysis transforms it into crystalline magnetite ($Fe_3O_4$)[26]. All $CoFe@FeO_x$ NPs are distributed uniformly on the carbon matrix, with particle size around 50 nm (Fig. S5). The $CoFe@FeO_x$ core@shell structure remains unchanged with the carburization temperature increasing from 240 to 500 °C (Fig. S6 and S7). The CoFe@C mainly contains Co, Fe, K, C, O, and N elements with Co and Fe account for 32.4% and 23.0% respectively (Fig. S8). The surface of CoFe@C and $CoFe@FeO_x$ are both dominated by C due to the carbon matrix and the surface iron state of $CoFe@FeO_x$ confirm the existence of $FeO_x$ shell. (Fig. S9).

The Fe and Co K-edge X-ray absorption near-edge structure (XANES) spectra of $CoFe@FeO_x$ lays in the middle of Fe foil, $Fe_3O_4$, and $Fe_xC$ references (Fig. 2a) and Co foil, $Co_2C$, and CoO references (Fig. 2b), indicating a mixture of CoFe alloy, iron/cobalt carbide and oxide phases. The Fourier transformation of Fe and Co k-edge extended X-ray absorption fine structure (EXAFS) spectrum exhibit distinct scattering peaks of Fe/Co−O, Fe/Co-C, and Fe-Co (Figs. 2c, d). Strong peaks of Fe-Co and weak ones of Fe/Co−O and Fe/Co-C suggest a large contribution from CoFe alloy. The experimental data of Fe and Co K-edge XANES and EXAFS spectra are in excellent agreement with the fitted data (Figure S10 and S11). Fe K edge wavelet transform extended X-ray absorption fine structure (WTEXAFS) present peaks of Fe-O, Fe-C, and Fe-Co, while that of Co K edge exhibits a main peak of Fe-Co (Figs. 2e, f, and S12). The coordination numbers (CNs) confirm the $CoFe@FeO_x$ NPs with a CoFe alloy core and a $FeO_x$ shell (Table S2).

The Mössbauer spectra of CoFe@C confirms the coexistence of CoFe alloy and $FeO_x$ species, with CoFe alloy and $FeO_x$ accounting for 96% and 4% respectively (Fig. 3a, Table S1). The Mössbauer spectra of $CoFe@FeO_x$ demonstrates a mixture of CoFe alloy, $FeO_x$, and $Fe_xC$ phases, with their content of 72%, 21%, and 7%, respectively (Fig. 3b). $Fe_xC$ denotes a special iron carbide with Bhf of 11.92 T. The phase composition of CoFe@C and $CoFe@FeO_x$ is also confirmed by X-ray diffraction (XRD) patterns (Fig. S13). The Mössbauer spectra of $FeO_x$ displays doublet peak and the fitted data of IS (0.26), QS (0.86), and Bhf (0.00) can be assigned to a type of iron oxide with superparamagnetic property and NP size below 10 nm. Combining with the HRTEM, XANES, and XRD of $CoFe@FeO_x$, 72% of CoFe alloy is ascribed to the core, 21% of $FeO_x$ to the shell, and 7% of $Fe_xC$ to the unoxidized shell of $CoFe@Fe_xC$ intermediate. The intermediate of $CoFe@Fe_xC$ is hard to be characterized directly due to its instability in air. The experimental evidence of this intermediate is the GC results of the exhaust during carburization. The exhaust contains methane, ethylene, ethane, propylene, propane, butene, butane, pentene, and pentane (Fig. S14), indicating the occurrence of catalysis and further indicating the existence of exposed $Fe_xC$ phase. $Fe_xC$ are considered as the active phases in Fischer-Tropsch synthesis (FTS). The carburization process is also a catalytic process of FTS[26]. The formed $Fe_xC$ shell can be passivated to form an outside $FeO_x$ shell[27,28]. An amorphous surface iron oxide layer has been noticed in the spent iron catalyst after reaction in thermal syngas, confirming the passivation of $Fe_xC$[29–32].

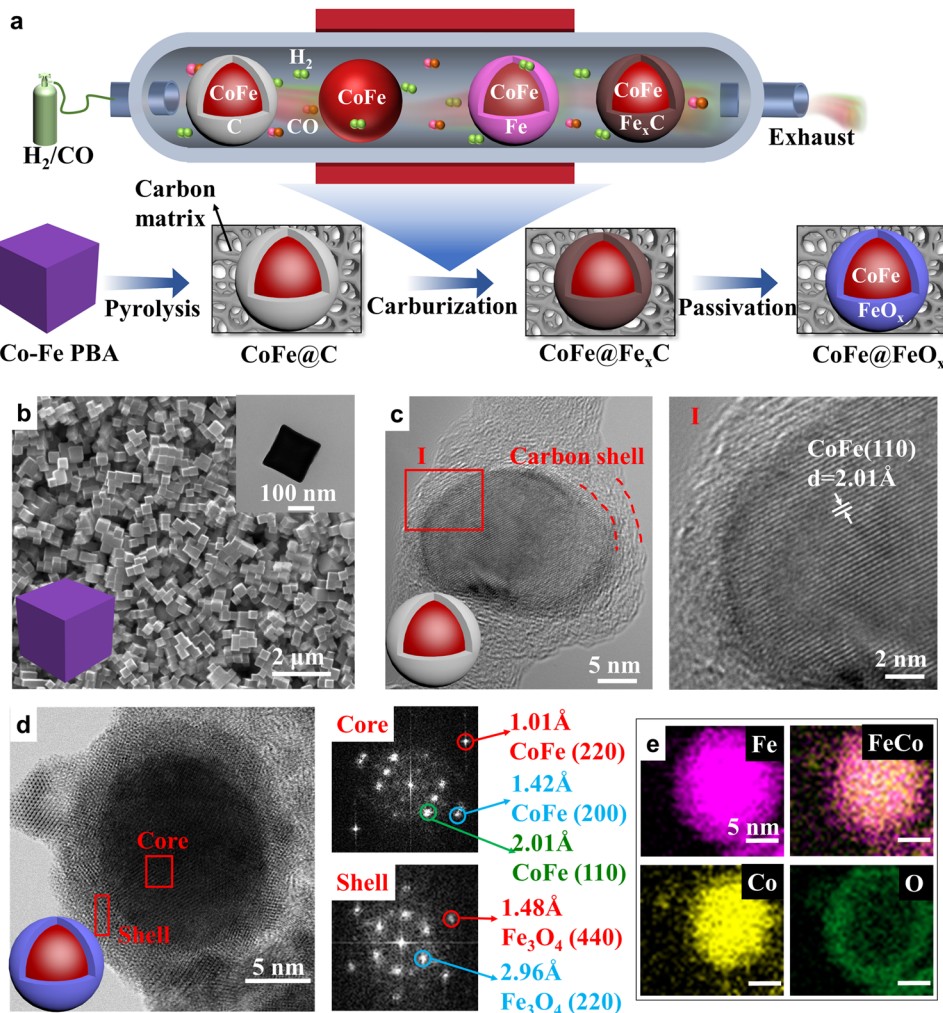

**Fig. 1 | Synthesis and characterization. a** Schematic diagram of the formation process of the CoFe@FeO$_x$. **b** SEM, TEM (inner) images, and scheme of Co-Fe PBA. **c** HRTEM image and the corresponding lattice fringes of the CoFe@C. **d** The STEM image, the FFT pattern, and **e** the corresponding elemental mapping images of the CoFe@FeO$_x$.

## Universal synthesis strategy for the MFe@FeO$_x$ (M = Co, Mn, Ni)

This solid-phase synthesis strategy via thermal syngas treatment can be generalized to synthesize the MnFe@FeO$_x$ and NiFe@FeO$_x$. The precursors of Mn-Fe PBA and Ni-Fe PBA are applied, which are obtained through substituting the cobalt ion of Co-Fe PBA by nickel and manganese ions (Fig. S15). The synthesized MnFe@FeO$_x$ NPs demonstrate a core@shell structure with 2.10 Å of lattice distance in the core and 2.40 Å of that in the shell, ascribing to the (101) and (311) planes of MnFe (PDF#03-0970) and Fe$_3$O$_4$ respectively (Figs. 3c, and d). The STEM and elemental mapping images also confirm the MnFe@FeO$_x$ core@shell structure (Fig. 3e). The NiFe@FeO$_x$ NPs exhibit a core@shell structure with 1.00 Å of lattice distance in the core and 1.50 Å of that in the shell, ascribing to the (311) and (440) planes of NiFe (PDF#38-0419) and Fe$_3$O$_4$ respectively (Figs. 3f, g, and h).

The FeO$_x$ shell thickness of CoFe@FeO$_x$ and NiFe@FeO$_x$ are 5 and 2 nm respectively. CoFe and NiFe alloy present body-centred cubic and face-centred cubic structure, respectively, and the segregation energy may be influenced by the crystal structure. The thinner shell thickness of NiFe@FeO$_x$ may come from the less difference of the segregation energy between Fe and Ni than that between Fe and Co, leading to less atoms exsolved from NiFe alloy to the surface than from CoFe alloy. This solid-phase synthesis strategy via thermal syngas treatment is universal and precise for the synthesis of the MFe@FeO$_x$ (M = Co, Mn, Ni). Furthermore, this strategy is supposed to be extended to the synthesis of more iron-based bimetallic NPs with excellent distribution, such as, CuFe, ZnFe, etc. The most important factor of this strategy is the exsolution of iron from iron-based alloy and the subsequent carburization. The facile operation and the abundant syngas indicate this solid-phase synthesis strategy is suitable for producing well-designed iron-based NPs in large scale.

## Theoretical formation mechanism of the CoFe@Fe$_x$C

Under a thermal syngas atmosphere, Fe atoms tend to exsolve from the bulk of CoFe alloy NPs to their surface, forming a Fe shell outside the CoFe alloy core. The formed Fe shell is further carburized into a Fe$_x$C shell by CO, constructing a CoFe@Fe$_x$C core@shell structure with CoFe alloy as the core and Fe$_x$C as the shell. Fe$_x$C presents a special iron carbide. The structural evolution is atomically shown in Fig. 4a. The exsolution and carburization processes may occur simultaneously.

It is interesting to observe that cobalt of CoFe@FeO$_x$ remains as alloy phase in the core rather than converts into cobalt carbides, since Co$_2$C phase is commonly observed in the syngas-treated CoMn materials[33]. This can be explained by the exsolution of Fe from CoFe alloy due to the different segregation energy of Co and Fe. In order to theoretically understand the exsolution process of Fe atoms from CoFe alloy, density functional theory (DFT) calculations are applied to

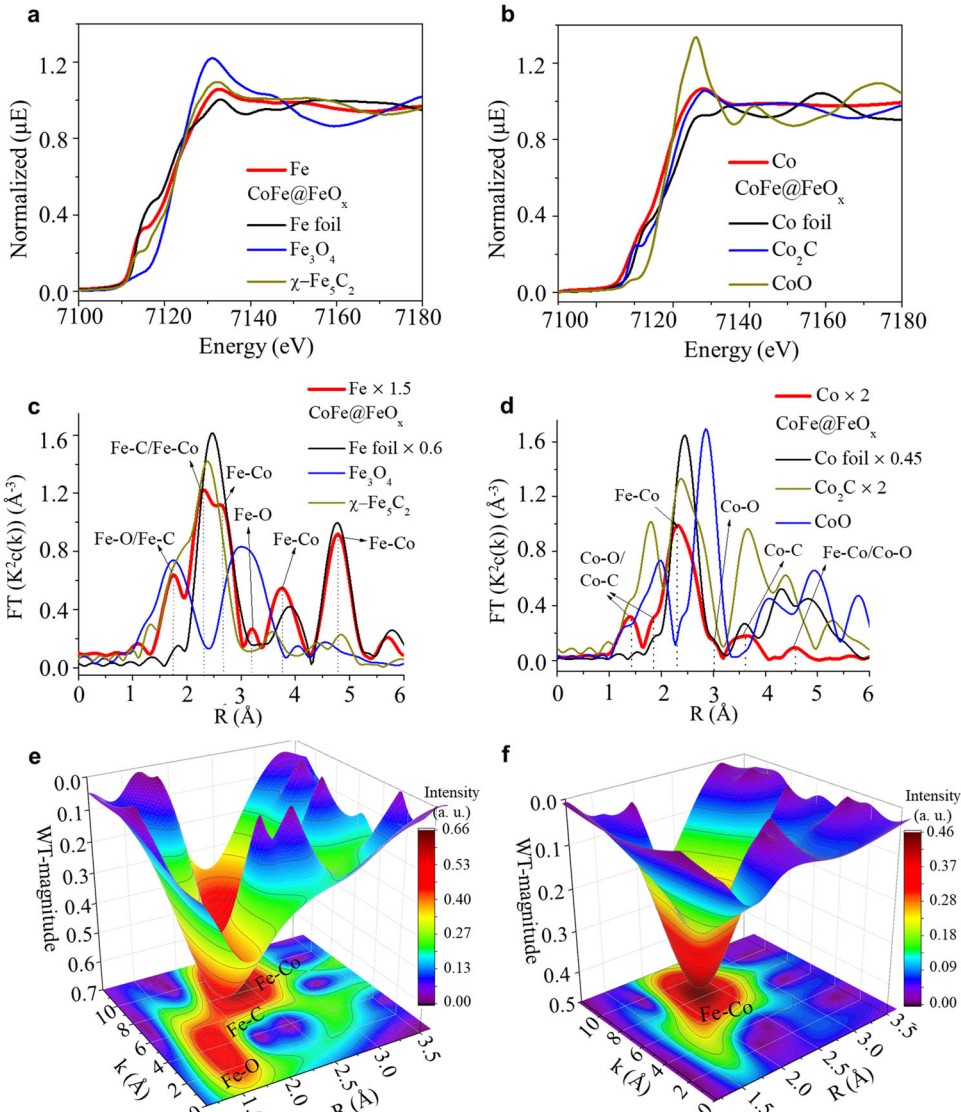

**Fig. 2 | The X-ray absorption spectroscopy of CoFe@FeO$_x$. a** Fe and **b** Co K edge XANES spectra, **c** Fe and **d** Co K edge EXAFS spectra of CoFe@FeO$_x$, as well as the Fe and Co references. **e** Fe and **f** Co K edge WTEXAFS of CoFe@FeO$_x$.

investigate the Fe/Co atom segregation energy and segregation pathway in the CoFe alloy (see details in the Supplementary Information). For Fe/Co atoms segregation energy in the CoFe alloy, the Co atom segregation from the bulk to the surface in the CoFe alloy is easier compared to the Fe atom segregation in the absence of CO adsorption (Fig. S16, S17, S18), however, in the presence of CO adsorption, the segregation of Fe atom becomes easy, and the segregation of Co atom is suppressed. Thus, Fe atom segregation in the CoFe alloy is preferred instead of Co atom segregation under the CO-rich atmosphere. Meanwhile, for Fe/Co atoms segregation pathway in the CoFe alloy with the presence of CO adsorption, as presented in Fig. S19 and S20, the reaction energies of step 1, step 2, step 3 and step 4 for Co atom segregation pathway are 1.79, −6.29, 14.00 and −7.04 eV, respectively. Hence the step 3 is strongly endothermic, and it is the rate-determining step. However, those of step 1, step 2, step 3, step 4 and step 5 for Fe atom segregation pathway are 1.99, 2.75, −4.65, 4.18 and −4.06 eV, respectively, thus, the step 4 is the rate-determining step, which is much lower than that of step 3 for Co atom segregation pathway (4.18 $vs.$ 14.00 eV) (Fig. 4b), suggesting that Fe atom segregation pathway is more energetically favorable than Co atom segregation pathway. The similar situation also occurs in the CoFe alloy with the absence of CO

adsorption. Furthermore, CO adsorption inhibits Co atom segregation in the CoFe alloy compared with the absence of CO adsorption (14.00 $vs.$ 4.87 eV). More importantly, Fe atom segregation with CO adsorption is more preferred to that without CO adsorption (4.18 $vs.$ 4.53 eV). Moreover, the metal Fe could be rapidly carburized to iron carbides under the FTS conditions (150-350 °C, 2-3 MPa)[34–36], which is attributed to CO adsorption. Thus, the easy segregation of Fe atom from the bulk to the surface results in the formation of Fe$_x$C shell outside the CoFe alloy core.

The driving force of the exsolution is the different segregation energy between Fe and Co in CoFe alloy (Fig. 4c). The segregation energy of Fe ($E_{seg-Fe}$) is lower than that of Co ($E_{seg-Co}$) under thermal syngas atmosphere, leading to the exsolution of Fe atoms and the formation of CoFe@Fe core@shell structure. The formed Fe shell can be carburized by CO and convert to Fe$_x$C shell, while the CoFe alloy core remains, due to the carburization ability of Fe ($\theta_{Fe}$) is much higher than that of CoFe alloy ($\theta_{CoFe}$) (Fig. 4d).

## Application of CoFe@FeO$_x$ for Li-S and Na-S batteries
The universal and facile synthesis of unique core@shell structure endows CoFe@FeO$_x$ material good application prospects in energy

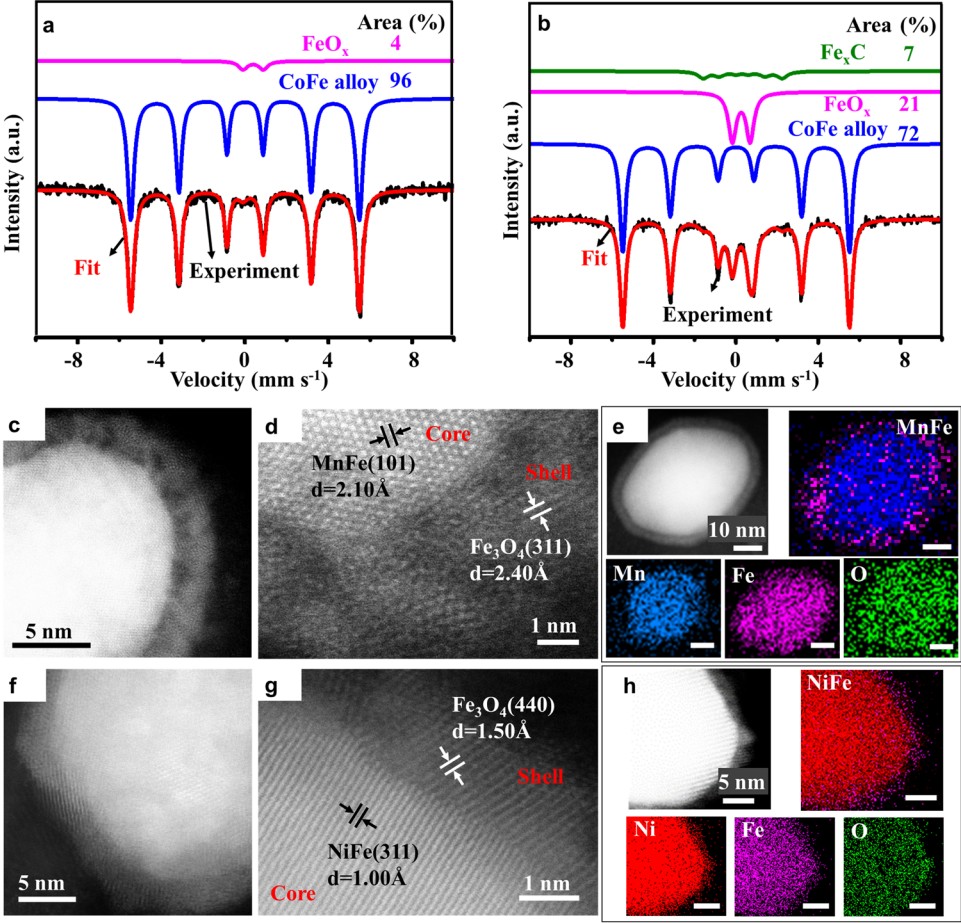

**Fig. 3 | Characterization of MFe@FeO$_x$ (M = Co, Ni, Mn).** The Mössbauer spectra of the **a** CoFe@C and **b** CoFe@FeO$_x$. **c** HRTEM image, **d** the corresponding lattice fringes, as well as **e** STEM image and the corresponding elemental mapping images of the MnFe@FeO$_x$. **f** HRTEM image, **g** the corresponding lattice fringes, as well as **h** STEM image and the corresponding elemental mapping images of the NiFe@FeO$_x$.

storage fields, *e.g.* as advanced modifying material of separators for Li-S batteries and Na-S batteries. Generally, traditional polypropylene (PP) separator has very limited effect for adsorbing the lithium polysulfides (LiPSs) in Li-S batteries, which may lead to the low utilization of S and poor electrochemical performance. Various conducting polymers and covalent–organic frameworks with strong chemical adsorptivity, or carbon matrices like carbon nanotubes or graphene with high conductivity have been employed as the modifying layer of commercial separators, and improving the utilization of S to some extent. However, most of these reported modifying layers hardly simultaneously possess strong adsorbability and high conductivity. The as-prepared CoFe@FeO$_x$ exhibits a bifunctional effect on regulating polysulfides as the separator coating layer for Li-S and Na-S batteries. In detail, the polar FeO$_x$ shell could effectively adsorb polysulfides in the surface, and the conductive CoFe core facilitates the conversion process of polysulfides, thus significantly suppressing the polysulfide shuttling effect. Commercial multiwalled carbon nanotubes (CNTs) and S powder composite is employed as cathode (Fig. S21) and the optical photographs of the modified PP separator (CoFe@FeO$_x$/PP) are shown in Fig. S22.

Systematic electrochemical tests for Li-S batteries are carried out in an environmental chamber with the temperature of 25 °C. The charge/discharge curves and cyclic voltammogram (CV) of Li-S batteries are shown in Fig. S23. Detailed performance comparisons of Li-S batteries with traditional PP, CoFe@C modified separator (CoFe@C/PP) and CoFe@FeO$_x$/PP are exhibited in Fig. S24. Moreover, the high areal S loading electrode (6.8 mg cm$^{-2}$) is fabricated (Fig. S25),

indicating good application prospect of CoFe@FeO$_x$ for boosting high energy density practical Li-S batteries. CV curves comparison with different scan rates are shown in Fig. S26. To directly compare the anchoring effect of LiPSs between the CoFe@FeO$_x$ and the CoFe@C, the shuttle current measurement is performed as shown in Fig. S27. The CoFe@FeO$_x$-based battery presents a lower shuttle current than that of the CoFe@C-modified one, indicating the stronger anchoring effect of the CoFe@FeO$_x$ for adsorbing LiPSs. The ability to accelerate the conversion process of LiPSs is tested by the Li$_2$S precipitation experiment (Fig. S28). The larger precipitation capacity with the CoFe@FeO$_x$/PP (122 mAh g$^{-1}$) demonstrates the good catalytic activity of the CoFe@FeO$_x$ to promote the conversion process of LiPSs. In addition, both the MnFe@FeO$_x$ and NiFe@FeO$_x$-based Li-S batteries exhibit good cycling stability as show in Fig. S29, suggesting MnFe@FeO$_x$ and NiFe@FeO$_x$ could also effectively restrain the shuttle of LiPSs.

In order to expand the application of the CoFe@FeO$_x$, the CoFe@FeO$_x$ modified commercial glass fiber (GF) separator (CoFe@FeO$_x$/GF) is further designed to regulate Na-S system, as shown in Fig. 5a. And all electrochemical tests for Na-S batteries are also performed in an environmental chamber with the temperature of 25 °C. The optical photographs and CNTs and S composite cathode are presented in Fig. S30. The CV curves of the CoFe@FeO$_x$-based Na-S battery with the scanning rate of 0.2 mV s$^{-1}$ (vs Na/Na$^+$) are exhibited in Fig. S31, which present typical oxidation peak (1.88 V) and reduction peak (1.16 V)[22]. The comparison of charge density differences is presented in Figs. 5b and c, which further demonstrates the strong interaction between Na$_2$S$_4$ and FeO$_x$. The optimized

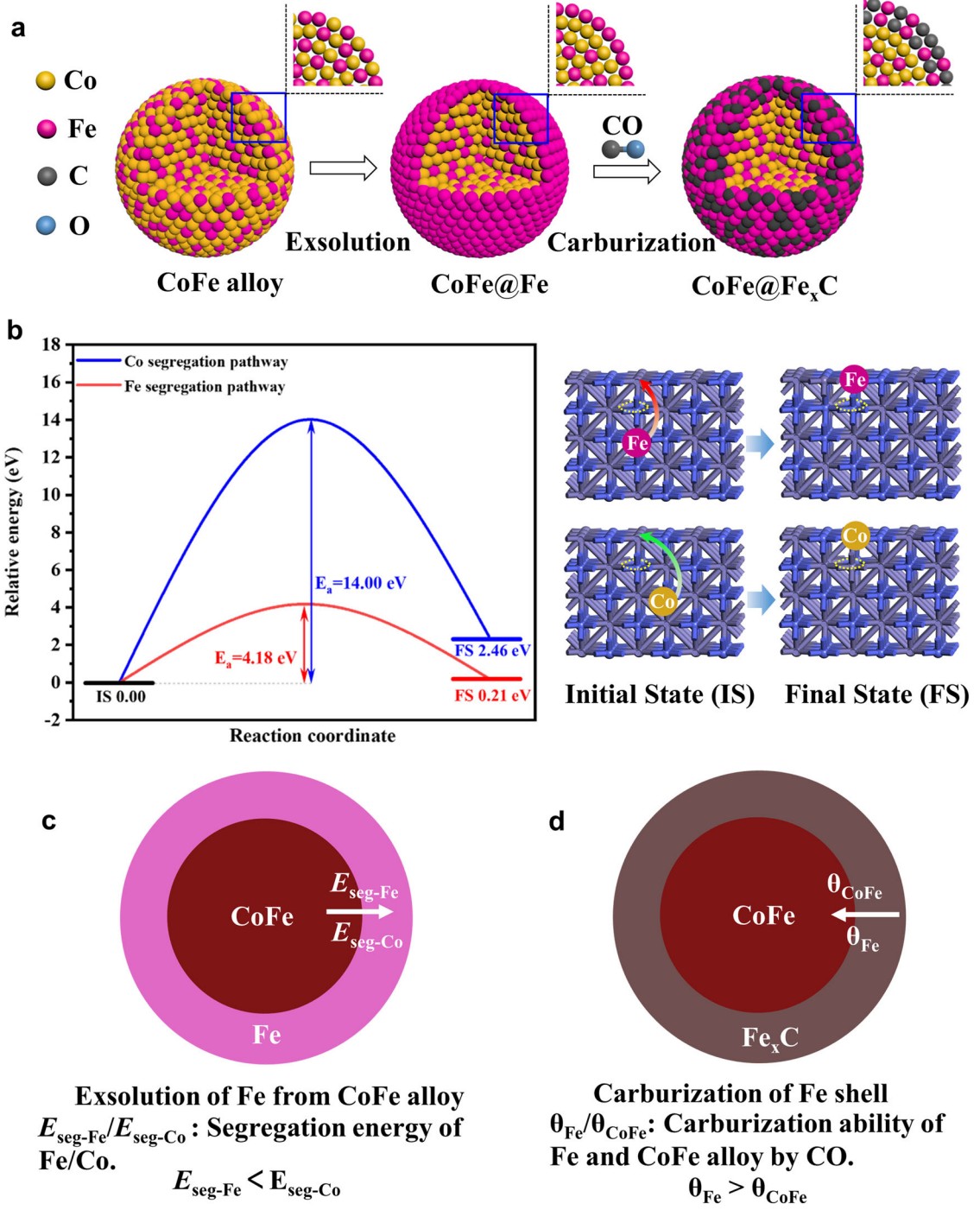

**Fig. 4 | CoFe@Fe$_x$C formation mechanism via Fe segregation from CoFe alloy.**
**a** Scheme of the atomic transformation from CoFe alloy to CoFe@Fe$_x$C. **b** The energy profile for the segregation pathway of Fe (red ball) and Co atom (blue ball) in the CoFe alloy with the Co vacancy (yellow circle) located in the 2$^{nd}$ layer in the presence of CO adsorption, and the blue and purple color balls are Co and Fe atoms, respectively. **c** The exsolution of Fe from CoFe alloy based on segregation energy. **d** CoFe@Fe$_x$C formed through the carburization of Fe shell by CO.

adsorption configuration between Na$_2$S$_4$ and FeO$_x$ or graphite were displayed in Figs. S32 and S33. The binding energy of Na$_2$S$_4$ with FeO$_x$ and graphite is -3.95 and -0.75 eV, respectively, indicating the CoFe@FeO$_x$ possesses more superior anchoring ability for polysulfides compared to the CoFe@C. The mechanism schematic of CoFe@FeO$_x$ enhancing the performance of Na-S batteries is illuminated in Fig. 5d. The core-shell CoFe@FeO$_x$ possesses a bifunctional effect on regulating sodium polysulfide (NaPSs), because the polar FeO$_x$ shell could effectively anchoring polysulfides in the surface and the conductive CoFe core further catalyzes the transformation

process of NaPSs, thus significantly enhancing the utilization of S and electrochemical performance of Na-S batteries. The cycle performances comparison at 0.2 A g$^{-1}$ are displayed in Fig. 5e. After 150 cycles, the battery with CoFe@FeO$_x$/GF can maintain a higher capacity of 772 mAh g$^{-1}$ than those with the CoFe@C/GF (396 mAh g$^{-1}$) and pure GF (49 mAh g$^{-1}$), indicating the CoFe@FeO$_x$ can effectively inhibit the shuttle of NaPSs and promote the conversion process. The rate performances are measured as shown in Fig. 5f and g. The Na-S battery with CoFe@FeO$_x$/GF could deliver the highest specific capacities than that of the CoFe@C modified one. Besides, the

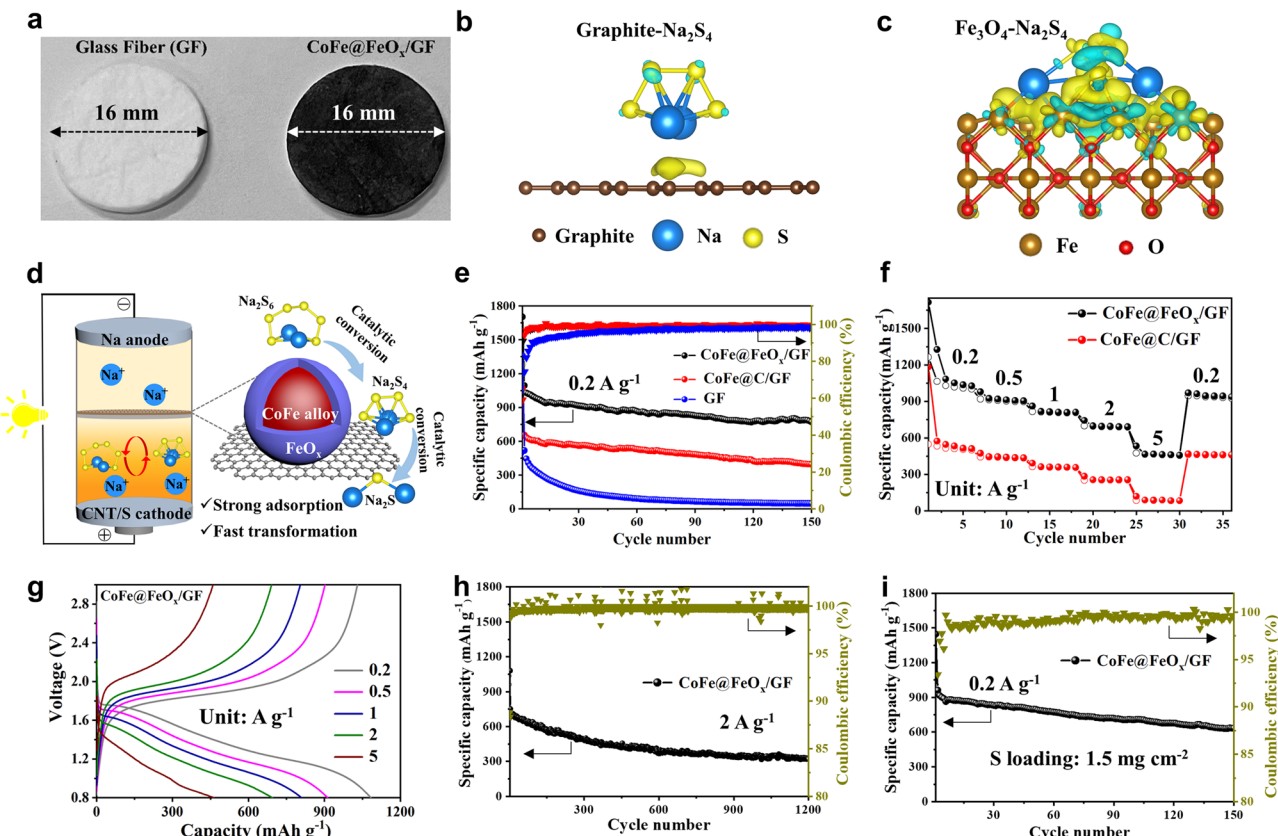

**Fig. 5 | Characterization and electrochemical performance with different separators. a** The optical photographs of the commercial Glass Fiber (GF) separator and CoFe@FeO$_x$/GF separator. The charge density differences of Na$_2$S$_4$ on the surface of **b** graphite and **c** Fe$_3$O$_4$. Cyan and yellow regions represent the decreased and increased electron density, respectively. The value of the isosurface is 0.002 electron bohr$^{-3}$. **d** The mechanism schematic diagram of the CoFe@FeO$_x$ catalyst in Na-S batteries. **e** Cycling performance comparison at 0.2 A g$^{-1}$. **f** Rate capability comparison at various current densities. **g** Typical charge/discharge curves at different current densities and **h** long-term cycling performance of CoFe@FeO$_x$/GF based Na-S battery at 2 A g$^{-1}$. **i** Cycling performance of CoFe@FeO$_x$/GF based battery at 0.2 A g$^{-1}$. The batteries in Fig. 5e-h are composed of Na metal anode and CNTs and S composite cathode with 50 wt% of sulfur loading. The battery in Fig. 5i is composed of Na metal anode and CNTs and S composite cathode with 70 wt% of sulfur loading. All electrochemical tests are performed in an environmental chamber with the temperature of 25 °C.

CoFe@FeO$_x$-based battery could maintain a high reversible capacity of 320 mAh g$^{-1}$ after 1200 cycles with nearly 100% Coulombic efficiency at 2 A g$^{-1}$ (Fig. 5h). More impressively, the Na-S battery with CoFe@FeO$_x$/GF can deliver a high reversible capacity of 935 mAh g$^{-1}$ with high S content (70 wt%) and keep a high capacity retention of 635 mAh g$^{-1}$ after 150 cycles at 0.2 A g$^{-1}$ (Fig. 5i). Even at a high current density of 2 A g$^{-1}$ (Fig. S34), the CoFe@FeO$_x$/GF based Na-S battery can display high reversible capacity of 306 mAh g$^{-1}$ after 600 cycles. The result indicates that CoFe@FeO$_x$ possess superior ability for suppressing the shuttle of NaPSs and facilitating their fast conversion process. The morphologies of Na anodes with commercial GF separator and CoFe@FeO$_x$/GF after three cycles at 0.2 A g$^{-1}$ are displayed in Fig. S35, demonstrating the CoFe@FeO$_x$/GF could inhibit the shuttle effect and protect the Na anode. Moreover, the MnFe@FeO$_x$/GF and NiFe@FeO$_x$/GF-based Na-S batteries could also deliver high capacities of 717 and 666 mAh g$^{-1}$ after 70 cycles at 0.2 A g$^{-1}$, respectively, indicating both the MnFe@FeO$_x$ and NiFe@FeO$_x$ can also suppress the shuttle effect of NaPSs and improve the utilization of S (Fig. S36). Thus, it can be concluded that the core-shell MFe@FeO$_x$ (M = Co, Mn, Ni) can effectively enhance the performances of Li-S and Na-S batteries because of their unique bifunctional effect on regulating polysulfides.

### Reaction mechanism for Na-S batteries

The electrochemical reaction mechanism of Na-S battery with CoFe@FeO$_x$/GF were further investigated by the ex-situ X-ray photoelectron spectroscopy (XPS). The peaks of S p$_{1/2}$ and S p$_{3/2}$ were captured in the original state as shown in Fig. S37a. When the battery was discharged to 1.6 V (Fig. S37b), soluble long-chain polysulfides (163.65 eV), thiosulfate and polythionate were detected[22]. And the peaks of short-chain Na$_2$S$_2$ species increased when discharged to 1.2 V (Fig. S37c). The peaks of Na$_2$S$_2$ and Na$_2$S obviously increased when the voltage was down to 0.8 V, indicating the complete conversion from S to Na$_2$S$_2$ and Na$_2$S (Fig. S37d). The S species exhibited reversible electrochemical behaviors during the charge process as shown in Fig. S37e. The S p$_{1/2}$ and S p$_{3/2}$ peaks were detected when charged to 3.0 V (Fig. S37f), suggesting the polysulfides were transformed into the original S$_8$ molecule[23].

## Discussion

A solid-phase synthetic strategy has been developed to precisely synthesize the well-dispersed CoFe@FeO$_x$ core@shell NPs via thermal syngas treatment. The CoFe@FeO$_x$ demonstrates a CoFe alloy core and a FeO$_x$ shell. According to DFT calculations, Fe atoms are preferentially exsolved from CoFe alloy bulk to the surface due to the lower Fe segregation energy than that of Co, and then the exsolved Fe shell is carburized into Fe$_x$C shell by CO, forming CoFe@Fe$_x$C structure, which is verified by the experimental results. The synthesis of the CoFe@FeO$_x$ starts from the pyrolysis of Co-Fe PBA. The CoFe@C with a CoFe alloy core and a carbon shell is obtained after the pyrolysis, which is well distributed on carbon matrix. The CoFe@Fe$_x$C intermediate is produced through treating CoFe@C under syngas

atmosphere at 240 °C for 24 h, whose surface $Fe_xC$ shell is prone to be passivated into $FeO_x$ by air at room temperature, generating the final $CoFe@FeO_x$ core@shell structure. The synthesized $CoFe@FeO_x$ NPs are uniformly distributed on the carbon matrix with ~50 nm of CoFe alloy core and 5 nm of $FeO_x$ shell. Analogously, the $NiFe@FeO_x$ and $MnFe@FeO_x$ are obtained through this strategy, demonstrating a universal strategy for the synthesis of $MFe@FeO_x$ (M = Co, Ni, Mn). The synthesized $CoFe@FeO_x$ demonstrate high performance as the modifying layer of commercial separators in Li-S and Na-S batteries. Benefiting from the adsorptive $FeO_x$ shell and conductive CoFe alloy core of $CoFe@FeO_x$, the polysulfides shuttling is well restrained and the conversion process of polysulfides is significantly enhanced. The Na-S battery can display a long cycle life of 1200 cycles with nearly 100% Coulombic efficiency. This strategy realizes the precise construction of complex core@shell metal NPs and their uniform distribution on supports simultaneously. The segregation energy and carburization ability of metals are applied to precisely control the spatial location of various iron-based phases at the nanoscale. Moreover, it also manifests interesting features of facile operation and solvent-free synthesis. This solid-phase synthesis strategy is not limited to synthesize iron-based NPs under thermal syngas, but could be developed into solid-phase synthetic systems to construct complex metal NPs.

## Online content

# Methods
## Chemicals
Cobalt nitrate hexahydrate ($Co(NO_3)_2 \cdot 6H_2O$), Nickel nitrate hexahydrate ($Ni(NO_3)_2 \cdot 6H_2O$), Manganese nitrate tetrahydrate ($Mn(NO_3)_2 \cdot 4H_2O$), potassium hexacyanoferrate (III) ($K_3Fe(CN)_6$), trisodium citrate dihydrate ($Na_3C_6H_5O_7 \cdot 2H_2O$) and sulfur powder were purchased from Sinopharm Chemical Reagent Co., Ltd. The commercial multiwalled carbon nanotubes (CNTs) and $Li_2S$ powder were purchased from Tokyo Chemical Industry (TCI) Shanghai. All chemical reagents were analytical grade and used without any treatment. Deionized water (DI) was used in all the above experiments. Washing was done with deionized water and reagent-grade ethanol.

## Material preparation
**Synthesis of Co-Fe PBA.** Co-Fe PBA precursor was prepared by a simple precipitation method, as reported by previous literatures[37]. Firstly, the solution A was obtained by dissolving 0.6 mmol of $Co(NO_3)_2 \cdot 6H_2O$ and 0.9 mmol of $Na_3C_6H_5O_7 \cdot 2H_2O$ into 20 ml of deionized water. Secondly, the solution B was obtained by adding 0.4 mmol of $K_3Fe(CN)_6$ into 20 ml of deionized water. Thirdly, the solution B was added slowly into the solution A under vigorously stirring. After continuously stirring for 2 min, the obtained precipitate was aged at room temperature for 24 h. Finally, the Co-Fe PBA precursor was obtained after centrifugation, washing with water and ethanol, and dry at 60 °C for overnight.

**Synthesis of CoFe@C.** The CoFe@C material was prepared by pyrolysis of Co-Fe PBA powder at 500 °C for 4 h with a heating rate of 3 °C $min^{-1}$ under a flow of $N_2$ atmosphere.

**Synthesis of CoFe@FeO$_x$.** The obtained CoFe@C (0.3 g, 20-40 mesh) was loaded into a fixed-bed reactor with an inner diameter of 10 mm and a bed length of 53 cm. The syngas treatment was carried out at 240 °C for 24 h, with the flowrate of $H_2$ and CO as 10 and 5 ml $min^{-1}$, respectively. The exhaust gas composition was analyzed by an online

gas chromatograph (GC) equipped with a thermal conductivity detector (TCD) and a flame ionization detector (FID). Products of methane, ethylene, ethane, propylene, propane, butene, butane, pentene, and pentane are detected by GC in the exhaust gas. After the syngas treatment, the temperature was decreased to 25 °C under syngas atmosphere, and then the syngas was changed to air for 1 h at 25 °C, with the flow rate of 10 ml $min^{-1}$. Finally, the $CoFe@FeO_x$ sample was obtained.

**Synthesis of NiFe@FeO$_x$ and MnFe@FeO$_x$.** The preparation of Ni-Fe PBA and Mn-Fe PBA is similar with that of Co-Fe PBA, except replacing the cobalt nitrate by nickel and manganese nitrate, respectively. The synthesis process of $NiFe@FeO_x$ and $MnFe@FeO_x$ are analogous to that of $CoFe@FeO_x$, which undergoes the pyrolysis of PBA, thermal syngas treatment, and air passivation.

**Synthesis of S composite cathode.** The commercial multiwalled CNTs and S powder composite was prepared by a melt-diffusion method. Typically, 30 mg of CNTs was mixed with 70 mg of sulfur powder with heat treatment at 155 °C for 24 h in an Ar-filled autoclave to gain the composite cathode with S content of 70 wt%. Similarly, 50 mg of CNTs was mixed with 50 mg of sulfur powder with heat treatment at 155 °C for 24 h in an Ar-filled autoclave to gain the composite cathode with S content of 50 wt%.

## Preparation of MFe@FeO$_x$ (M = Co, Mn, Ni) modified separators
The $CoFe@FeO_x$/PP ($CoFe@C$/PP, $MnFe@FeO_x$/PP or $NiFe@FeO_x$/PP) separator was prepared by mixing 95 wt% of $CoFe@FeO_x$ ($CoFe@C$, $MnFe@FeO_x$ or $NiFe@FeO_x$) with 5 wt% Sodium Carboxymethyl Cellulose (CMC) binder in deionized water to cast on one side of commercial PP separator (Celgard 2400). The $CoFe@FeO_x$/PP ($CoFe@C$/PP, $MnFe@FeO_x$/PP or $NiFe@FeO_x$/PP) separator was obtained after vacuum-dried at 70 °C for all night, followed by cutting into disks with 16 mm in diameter. Similarly, the $CoFe@FeO_x$/GF ($CoFe@C$/GF, $MnFe@FeO_x$/GF or $NiFe@FeO_x$/GF) separator was prepared with the same process except that replacing PP separator with commercial glass fiber separator (Whatman GF/A 1823-070).

## Li$_2$S precipitation experiment
The $Li_2S$ precipitation experiment was tested on the 2032-type coin cells assembled with the tested electrode, lithium foil and PP separator on the Autolab PGSTAT 302 N workstation. $Li_2S_8$ electrolyte (0.2 mol $L^{-1}$) was prepared by mixing sulfur with $Li_2S$ at a molar ratio of 7: 1 in tetraglyme followed by vigorous mixing for all night. Commercial carbon papers (Guangdong Canrd New Energy Technology Co. Ltd.) were used as current collectors to load the well-mixed slurry composed of 70 wt% $CoFe@FeO_x$ (or $CoFe@C$), 20 wt% Super P and 10 wt% CMC. The tested electrodes (loading mass: -1.5 mg $cm^{-2}$) were obtained after drying at 80 °C for 12 h. 30 mL $Li_2S_8$ was dropped onto the tested electrodes during the cell assembly process. Cells were first discharged galvanostatically at 0.112 mA to 2.09 V and then discharged potentiostatically at 2.05 V for $Li_2S$ nucleation and growth. The potentiostatic discharge was terminated when the current was below $10^{-5}$ A. Based on Faraday's law, the energy was collected to evaluate the nucleation/growth rate of $Li_2S$ on the tested electrodes.

## Characterization
XRD was performed on a Rigaku D/Max2500PC diffractometer with 2θ range of 5-80 °. The Cu Kα radiation was used with the voltage of 40 kV. TEM images were obtained on FEI Tecnai $G^2$ Spirit microscope with 120 kV. HRTEM images were operated on a FEI Tecnai $G^2$ F30S-Twin microscope with 300 kV. HAADF-STEM and EELS were operated on a JEM-ARM200F thermal-field emission microscope. SEM images were measured on a FEI Quanta 200F device. XPS was performed on a

KRATOS Axis Ultra$^{DLD}$ spectrometer. An Al Kα X-ray radiation source (1486.6 eV) and charge compensation gun were used. The C 1 s peak (284.60 eV) was used to perform the charge correction. Inductively coupled plasma optical emission spectrometry (ICP-OES) was measured on an ICPS-8100. For sample preparation, calcination treatment was performed at 600 °C in air (carbon species removement), and then the obtained sample was dissolved in an acidic mixture of $HNO_3$ and HCl. The element content of O, N, and H was determined by an element analyzer of EMGA-930. The element content of C was determined by an element analyzer of EMIA-8100.

The RT $^{57}$Fe Mössbauer spectra were acquired from a proportional counter and a Topologic 500 A spectrometer. A $^{57}$Co (Rh), moving with a constant acceleration mode, was applied as the γ-ray radioactive source. A standard α-Fe foil was used as a reference. The spectra were fitted on the base of Lorentzian adsorption curves using MossWinn 3.0i computer program. The derived hyperfine parameters of isomer shift (IS), quadruple splitting (QS), and magnetic hyperfine filed (H), were applied for component identification. The phase content was confirmed on the base of the areas of the adsorption peaks, assuming the iron nuclei for all samples possess the same probability of adsorption of γ photons. The X-ray absorption spectra including XANES and EXAFS of the samples at K-edge were collected at the Beamline of TLS07A1 in National Synchrotron Radiation Research Center (NSRRC), Taiwan, where 1.5 GeV a pair of channel-cut Si (111) crystals was used in the monochromator.

### Computational details

**Calculation methods.** All spin-polarized DFT calculations were implemented in the Vienna Ab initio Simulation Package (VASP) software[38]. The generalized gradient approximation (GGA)[39] together with the Perdew-Burke-Ernzerhof (PBE) functional[40] was performed to describe the electronic exchange-correlation functions. The electronic wave functions were expanded using a kinetic energy cutoff of 400 eV. The projector-augmented plane wave (PAW) was carried out to perform the electron–ion interactions[41,42]. Surface Brillouin-zone corresponds to a 3×3×1 $k$-point grid. The optimization convergence accuracy of the force and energy was less than 0.03 eV Å$^{-1}$ and 1×10$^{-5}$ eV, respectively.

**Calculation models.** CoFe alloy was modeled using atomic layers oriented along the (110) plane. For CoFe(110) surface, a five-layer $p$(3×2) supercell is constructed, the Fe and Co atoms of top four layers and the adsorbates are allowed to relax.

**The surface segregation and segregation energy.** Previous studies showed that the adsorption of CO can alter the surface segregation of metal materials[43], so the segregation behaviors of Fe and Co atoms for the CoFe alloy in the absence and presence of CO were examined. Meanwhile, the segregation energy ($E_{seg}$) is defined as the energy required for a single Fe or Co atom to move from the bulk to the surface layer, which can be calculated using the following equation[44–46]:

$$E_{seg-n(n=1-3)} = E_{surf\,n(n=1-3)} - E_{surf4} \tag{1}$$

Where $E_{surf\,n(n=1-3)}$ represents the energy of Fe or Co atom located into the $n^{th}$ layer of CoFe alloy; $E_{surf\,4}$ represents the energy of Fe or Co atom located into the 4$^{th}$ layer of CoFe alloy. $E_{seg-n(n=1-3)}$ represents the segregation energy, namely, the energy of Fe or Co atom located into the 4$^{th}$ layer of CoFe alloy being transferred to the $n^{th}$ layer. The more negative value of $E_{surf\,n(n=1-3)}$ means that a Fe or Co atom is easier to move from the bulk to the surface layer.

**The segregation pathway of Fe or Co in the CoFe alloy.** Usually, the alloy segregation occurs with the exchanges between the metal atom

and surface/subsurface vacancies[43,47,48], for example, DFT studies by Zhang et al.[37]. fully researched the segregation pathway of Ni atom in Au catalyst, specifically, an Au vacancy was set initially at the second atomic layer and then possible segregation pathway including Ni atom near the vacancy to alter its atomic position with a surface Au atom through a series Au/Ni-vacancy exchange are considered. In this study, the similar Co/Fe segregation pathway in the CoFe alloy is examined using DFT calculations; meanwhile, the same alloy surface used in the segregation energy calculation was adopted as the model surface. A Co or Fe vacancy was initially set in the 2$^{nd}$ layer. Further, a possible pathway is proposed for the 3$^{rd}$ layer Fe or Co atom near the vacancy to change the atomic position with a surface Fe or Co atom through a series of Fe/Co-vacancy exchange steps (see details in the Supplementary Information).

### Electrochemical investigation

2032-type coin cells are assembled to evaluate the electrochemical performances of Li-S batteries and Na-S batteries in an Ar-filled glove box ($O_2 < 0.01$ ppm, $H_2O < 0.01$ ppm). The cathode materials are prepared by blending 80 wt% active materials, 10 wt% carbon black and 10 wt% polyvinylidenedifluoride (PVDF) and pasted onto an aluminum foil (thickness: 50 μm). A Na foil (the diameter is 10 mm, the thickness is 300-400 μm) is used as the counter electrode. The areal S loading of the common cathode is 1.5 mg cm$^{-2}$. The average mass loading of CoFe@FeO$_x$ or CoFe@C on the PP separator (thickness: 25 μm) is controlled to be around 0.25 mg cm$^{-2}$. And the average mass loading of CoFe@FeO$_x$ or CoFe@C on glass fiber separator (thickness: 300 μm) is controlled to be around 0.4 mg cm$^{-2}$. The diameter of PP separator or glass fiber separator is 16 mm. 20 μL of electrolyte is used in Li-S batteries, which is composed of 1 M lithium bis(trifluoromethanesulfonyl)imide (LiTFSI) in a solvent mixture of 1,3-dioxolane (DOL) and dimethoxymethane (DME) (1:1 by volume) with 1 wt% LiNO$_3$. And a solution of 1 M sodium bis(trifluoromethylsulfonyl)imide (NaTFSI) in propylene carbonate (PC) /fluoroethylene carbonate (FEC) (1:1 by volume, 60 μL) is utilized as the electrolyte for Na-S cells. The galvanostatic charge-discharge tests were conducted on the Neware BTS-610 instrument. The cyclic voltammetry measurements and electrochemical impedance spectrum (EIS) measurements were obtained on the CHI 660D workstation. All tests of cells are carried out in an environmental chamber with the temperature of 25 °C.

### Data availability

The main data supporting the findings of this study are available within the main text, the Supplementary Information file, and the Source Data files. Additional raw data are available at https://doi.org/10.6084/m9.figshare.24426877. Source data are provided with this paper.

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

## Acknowledgements

This work was supported by the National Key R&D Program of China (2021YFA1501900, 2022YFA1504100, 2022YFA1504500), National Natural Science Foundation of China (Nos. 51925207, 52372239, U1910210, 52161145101, 52102322, 62227815, and 22108270), the National Synchrotron Radiation Laboratory (KY2060000173), the Joint Fund of the Yulin University and the Dalian National Laboratory for Clean Energy (Grant. YLU-DNL Fund 2021002), the Fundamental Research Funds for the Central Universities (WK2060000055, WK2400000004). We acknowledge Q. Jiang, W. Liu, Y. Zhao, and R. Han (Dalian Institute of Chemical Physics, Chinese Academy of Sciences) for valuable discussions on the HRTEM images.

## Author contributions

J.L., Y.Yu, and R.Z. conceived the idea for the project and supervised the project. Y.C. and H.L. performed the material synthesis and

characterization. Y.Yao, Y.J., and L.W. performed the electrochemical test. W.Z., B.W., and R.Z. performed the theoretical calculations. J.Z. performed X-ray absorption spectra experiments. Y.C. and Y.Yao wrote the manuscript. All authors discussed and commented on the manuscript.

## Competing interests

The authors declare no competing interests.
