## [Peer Review File · Nature Communications]

Precision solid synthesis of CoFe@FeOx nanoparticles for efficient polysulfide regulation in Li-S and Na-S batteriesREVIEWER COMMENTS

Reviewer #1 (Remarks to the Author):

In the manuscript 'Precision solid synthesis of CoFe@FeOx nanoparticles for regulating Li/Na-S batteries' Chen et al. presented a universal synthesis method of CoFe@FeOx by thermal syngas decomposition method and its application in Li-S or Na-S batteries. The formation of CoFe@FeOx is explained by the computational and experimental approach. However, the manuscript lacks novelty and many following questions need to be addressed. The manuscript in its present form will not attract readers.

1. Can the authors explain Mossbauer spectroscopy in more detail? How exactly are the CoFe alloy and FeOx phases confirmed through this analysis?
2. Could the authors provide details such as the cathode source and sulfur loading for the LiS battery, the performance of which is shown in Figures S19 and S20?
3. How is the separator modified with CoFe@FeOx? Is the modification done on only one side of the separator or both sides?
4. How does CoFe@FeOx contribute to minimizing the polysulfide shuttling effect? What would happen with only CoFe/PP or FeOx/PP?
5. How were the Li₂S precipitation experiments carried out?
6. The y-axis units in Figure S22 A, B, and Figure S26 appear incorrect.
7. There is confusion regarding the sulfur loading. Somewhere authors claim 70 %, somewhere, 50 %. Authors should clearly mention the sulfur loading in every electrochemical measurement.
8. Authors claim shell is composed of graphitized and amorphous carbon. How have the authors confirmed that? What is the role of graphitized and amorphous carbon?
9. Authors have synthesized MnFe@FeOx and NiFe@FeOx; however, their performance is not shown for Li S or Na S batteries. Why so? What advantage do CoFe@FeOx has over MnFe@FeOx or NiFe@FeOx

Reviewer #2 (Remarks to the Author):

This manuscript developed a solid synthesis strategy to precisely synthesize CoFe@FeOx core@shell NPs. The introduction elaborated in detail the development of the traditional metal NPs synthesis strategies and the advantages of the solid synthesis approaches. The synthesis process of CoFe@FeOx are presented carefully together with the structure and composition are characterized clearly. The synthesis mechanism is investigated according to the experimental results. This solid synthesis strategy is universal for the preparation of MFe@FeOx (M=Co, Mn, Ni). The synthesized CoFe@FeOx demonstrate high performance as the modifying layer of commercial separators in Li/Na-S batteries. This manuscript provides a novel perspective to synthesize complex metal NPs under thermal gas atmosphere, different from liquid environment of the traditional wet chemical methods. This solid synthesis strategy may enrich the current synthetic method of complicated structured metal NPs. Overall this manuscript is well written and I recommend to accept it after addressing following comments:

1. It seems that the theoretical formation mechanism of the CoFe@FexC confirms the experimental synthesis of the CoFe@FeOx. It is more reasonable to start with the synthesis and characterization of the CoFe@FeOx, and then follow with the theoretical formation mechanism. The Co₂C phases are commonly observed in the syngas treated CoMn

materials (Nature 2016, 538, 84-88.) For the CoFe alloy, the cobalt tends to be remained as alloy rather than converted into carbides. This interesting phenomenon is better to be explained with the physicochemical properties of Co and Fe.

2. The universal synthesis strategy section, NiFe@FeOx and NiFe@FeOx are also well characterized. Beside cobalt, what is the reason for the selection of Mn and Ni? Are they share some similar properties? What is the most important factor to affect the formation of MFe@FeOx core@shell structure? Can this synthesis strategies be extended to more bimetallic iron-based alloys? The forward-looking prospect may benefit the understanding of this synthesis strategy.

3. From Figure 2D and 4F, the FeOx shell thickness of CoFe@FeOx and NiFe@FeOx are 5 and 2 nm respectively. What is the reason for this difference? The synthesis and characterization section mainly focuses on the description of the morphology, the structure, and composition of CoFe@FeOx. The scientific reasons of these superficial phenomenon should be emphasized.

4. The synthesized CoFe@FeOx demonstrates high performance as the modifying layer of commercial separators in Li/Na-S batteries. The performance data of CoFe@FeOx is explained in detail. Compared with the separator materials in the current literatures, what is the advantages of the CoFe@FeOx? What is the inspiration for future research? Furthermore, what about the performance of NiFe@FeOx and MnFe@FeOx in Li/Na-S batteries? Maybe some rules or trends of these bimetallic iron-based materials can be summarized according to their performance in batteries.

5. Some minor errors should be corrected. Page 2, line 25, a space should be added between CoFe@FeOx and core@shell. Page 3, line 33, Prussian blue analog is better to be replaced by Prussian blue analogue. Page 5, line 37, 0.2 mV/s should be revised as 0.2 mV s⁻¹. The text font in Figure 5 is too small. The manuscript should be fully checked carefully.

Reviewer #3 (Remarks to the Author):

Chen et al. have developed a universal thermal syngas treatment method for synthesizing MFe@FeOx (M=Co, Ni, Mn). The obtained CoFe@FeOx nanoparticles have been thoroughly characterized. Furthermore, these CoFe@FeOx nanoparticles have been utilized as a separator coating layer in Li/Na-S batteries. This research work is highly intriguing and well-organized. The reviewer recommends publication of this work after making minor revisions.

1. The article proposed the synthesis mechanism of CoFe@FeOx according to the theoretical calculation. The segregation pathway of cobalt atom included four steps and that of iron atom is five steps. The segregation occurs with the exchanges between the cobalt/iron atom and surface/subsurface vacancies. I wonder where the initial vacancies come from. The theoretical calculation is supposed to verify the experiment results. For the experimentally prepared CoFe alloy NPs, do the vacancies appear in the CoFe crystal structure? I consider, whether this theoretical model can truly reflect the transformation of the CoFe crystal.

2. The preparation process of CoFe@FeOx is presented detailedly in Figure 1A. The structure and phase composition of CoFe@C is exhibited in Figure 2C and Figure 4A, as well as that of CoFe@FeOx is displayed in Figure 2D, Figure 2E, Figure 3, Figure 4B. However, the intermediate of CoFe@FexC show less characterization information. Is there some related experimental evidence about this intermediate.

3. It is better to make a brief introduction to the development status of the current separator

materials for Li/Na-S batteries. Whether literatures reported before about the application of core@shell metal NPs in batteries. What are the unique advantages of the CoFe@FeOx core@shell structure?

4. Cobalt atoms are yellow and iron atoms are pink in Figure 1A, while cobalt atoms are green together with iron atoms are red in Figure 1B. The cobalt and iron atom color should be consistent with each other. The FexC shell of CoFe@FexC scheme is blue in Figure 1D, while it is brown in Figure 2A. The FeOx shell is blue in Figure 2A and 2D, while it is brown in Figure 5D. The color of FexC and FeOx shell in the scheme should be consistent with each other.

Reviewer #4 (Remarks to the Author):

The authors reported a solid synthesis strategy route for the synthesis of complex metal NPs by using syngas as medium, providing a totally different synthetic route from the traditional liquid-phase medium. Such gas-phase medium route allow them to prepare CoFe@FeOx under syngas at 240 °C, this temperature is remarkably lower than the traditional CVD method. The authors demonstrated the carbon atoms of the syngas could be uniquely introduced into the metal crystal to form metal carbides intermediates, which are used to regulate the final metal structure. This method brings new concepts for the structural design of metal NPs. While the concept of achieving precise NP structure design and uniform dispersion is intriguing, several issues can be raised regarding the presented work. Below please find the comments.

1. The authors emphasize the comparison between the traditional wet chemistry methods and this thermal gas treatment method in the introduction. The current development of thermal gas treatment method for preparing materials is suggested to added.
2. Whether the structure of CoFe@FeOx is influenced by the syngas treatment temperatures. For example, when the treatment temperature increases to 300 °C, or even 500 °C, what will happen for the metal structure?
3. Co and Fe are both excellent syngas conversion catalysts. The thermal syngas treatment can also be considered as a catalytic reaction process. Does the composition of the exhaust gas in Figure 2A be monitored during the preparation process? This may help to understand the CoFe structure evolution.
4. Syngas is a type of largely produced chemical raw gas. This thermal syngas treatment methods present facile operation, indicating potential large-scale production. The authors are suggested to make a reasonable perspective about this method.
5. In introduction section, it likely lacks sufficient information regarding the specific limitations of wet chemistry methods in achieving both NP structure design and uniform dispersion simultaneously. It would be beneficial to elaborate on these limitations to provide a clearer context for the proposed solid synthesis strategy.
6. The methodology proposed in this work raises concerns about the exsolution and carburization processes. The selective exsolution of Fe atoms from the CoFe alloy bulk to the surface and their subsequent carburization into a FexC shell under thermal syngas atmosphere may benefit from a more detailed explanation. Additionally, the authors are suggested to provide information on the reaction conditions and detail process, which are crucial for reproducibility and assessing the feasibility of the proposed method.
7. It is essential or desirable to include experimental data or theoretical explanations that demonstrate the generalizability of the solid synthesis strategy for other metal compositions.

8. Additionally, the bifunctional effect of the CoFe@FeOx NPs on regulating polysulfides as a separator coating layer for Li/Na-S batteries was studied, however, the authors did not well clarify why CoFe@FeOx/GF can significantly enhance the specific capacity of Na-S batteries.

Response to Reviewers' Comments

Dear Editor,

We would like to thank the reviewers for their insightful comments and great efforts towards improving our manuscript. Following Reviewers' suggestions, we revised the manuscript. Please find below our replies to the comments of you and reviewers and description of revisions.

Reviewer #1

In the manuscript 'Precision solid synthesis of CoFe@FeO_x nanoparticles for regulating Li/Na-S batteries' Chen et al. presented a universal synthesis method of CoFe@FeO_x by thermal syngas decomposition method and its application in Li-S or Na-S batteries. The formation of CoFe@FeO_x is explained by the computational and experimental approach. However, the manuscript lacks novelty and many following questions need to be addressed. The manuscript in its present form will not attract readers.

Comment 1:

Can the authors explain Mossbauer spectroscopy in more detail? How exactly are the CoFe alloy and FeO_x phases confirmed through this analysis?

Response 1:

We would like to thank the reviewer's valuable comments. The experimental Mössbauer spectra of CoFe@C exhibits symmetrical six peaks, which is fitted well with 96 wt% CoFe alloy and 4 wt% FeO_x (Figure R1A). The IS (0.011), QS (0.00), and Bhf (34.0) of the fitted data belong to the typical iron-based alloy (Sci. Rep-UK, 2016, 6, 26184.), hence assigning to CoFe alloy (Table R1). The IS (0.26), QS (0.86), and Bhf (0.00) of the fitted data are ascribing to a kind of iron oxides (Table R1). The doublet peak of FeO_x presents its superparamagnetic property, indicating its particle size is below 10 nm (Figure R1B). For CoFe@FeO_x, the shell thickness of FeO_x is about 5 nm. Hence, the fitted Mössbauer spectra with doublet peak is assigned to the FeO_x shell. Furthermore, the CoFe alloy and FeO_x phases are also confirmed by XRD, HRTEM, and X-ray adsorption spectroscopy.

Figure R1 The Mössbauer spectra of the CoFe@C (A) and CoFe@FeO_x (B).

Table R1 Mössbauer parameters of the CoFe@C and CoFe@FeO_x

Samples	Assignment	IS (mm s ⁻¹)	QS (mm s ⁻¹)	Bhf (T)	Area (%)
CoFe@C	CoFe alloy	0.011	0.00	34.0	96
	FeO _x	0.40	0.97	0.0	4
CoFe@FeO _x	CoFe alloy	0.012	0	34.19	72
	FeO _x	0.26	0.86	0	21
	Fe _x C	0.32	0	11.92	7

Following the reviewer's comments, we made revision as below and in the revised manuscript:

"The Mössbauer spectra of FeO_x displays doublet peak and the fitted data of IS (0.26), QS (0.86), and Bhf (0.00) can be assigned to a type of iron oxide with superparamagnetic property and NP size below 10 nm. Combining with the HRTEM, XANES, and XRD of CoFe@FeO_x, 72 wt% of CoFe alloy is ascribed to the core, 21 wt% of FeO_x to the shell, and 7 wt% of Fe_xC to the unoxidized shell of CoFe@Fe_xC intermediate." (Page 3, line 43)

Comment 2:

Could the authors provide details such as the cathode source and sulfur loading for the LiS battery, the performance of which is shown in Figures S19 and S20?

Response 2:

Thanks for the reviewer's valuable comments. We have provided the details including the cathode source and sulfur loading in the Methods part. The commercial multiwalled carbon nanotubes (CNTs) were purchased from Tokyo Chemical Industry (TCI) Shanghai, and sulfur powder were purchased from Sinopharm Chemical Reagent Co., Ltd., and the average sulfur loading for the Li-S battery is around 1.5 mg cm⁻².

Following the reviewer's comments, we made revision as below and in the revised manuscript:

*"**Chemicals.** Cobalt nitrate hexahydrate (Co(NO₃)₂·6H₂O), Nickel nitrate hexahydrate (Ni(NO₃)₂·6H₂O), Manganese nitrate tetrahydrate (Mn(NO₃)₂·4H₂O), potassium hexacyanoferrate (III) (K₃Fe(CN)₆), trisodium citrate dihydrate (Na₃C₆H₅O₇·2H₂O) and sulfur powder were purchased from Sinopharm Chemical Reagent Co., Ltd. The commercial multiwalled carbon nanotubes (CNTs) and Li₂S powder were purchased from Tokyo Chemical Industry (TCI) Shanghai. All chemical reagents were analytical grade and used without any treatment. Deionized water (DI) was used in all the above experiments. Washing was done with deionized water and reagent grade ethanol."* (Page 16, line 2)

*"**Electrochemical investigation.** 2032-type coin cells are assembled to evaluate the electrochemical performances of Li-S batteries and RT Na-S batteries. The cathode materials are prepared by blending 80 wt% active materials, 10 wt% carbon black and 10 wt% polyvinylidenedifluoride (PVDF) and pasted onto an aluminum foil. The areal S loading of the common cathode is 1.5 mg*

cm⁻². The average mass loading of CoFe@FeO_x or CoFe@C on the PP separator is controlled to be around 0.25 mg cm⁻². And the average mass loading of CoFe@FeO_x or CoFe@C on glass fiber separator is controlled to be around 0.4 mg cm⁻². The electrolyte for Li-S batteries is composed of 1 M lithium bis(trifluoromethanesulfonyl)imide (LiTFSI) in a solvent mixture of 1,3-dioxolane (DOL) and dimethoxymethane (DME) (1:1 in volume) with 1 wt% LiNO₃. And the electrolyte for RT Na-S batteries consists of 1 M NaTFSI in Fluoroethylene carbonate (FEC): Polycarbonate (PC) (v/v 1:1). The galvanostatic charge-discharge tests were conducted on the Neware BTS-610 instrument. The cyclic voltammetry (CV) measurements and electrochemical impedance spectrum (EIS) measurements were obtained on the CHI 660D workstation.” (Page 19, line 2)

Comment 3:

How is the separator modified with CoFe@FeO_x? Is the modification done on only one side of the separator or both sides?

Response 3:

Thanks for the reviewer’s valuable comments. We have provided the preparation of CoFe@FeO_x or CoFe@C modified separators in the Methods part. And it should be noted that the modification layer is done on only one side of the separator.

Following the reviewer’s comments, we made revision as below and in the revised manuscript:

“Preparation of MFe@FeO_x (M=Co, Mn, Ni) modified separators

The CoFe@FeO_x/PP (CoFe@C/PP, MnFe@FeO_x/PP or NiFe@FeO_x/PP) separator was prepared by mixing 95 wt% of CoFe@FeO_x (CoFe@C, MnFe@FeO_x or NiFe@FeO_x) with 5 wt% Sodium Carboxymethyl Cellulose (CMC) binder in deionized water to cast on one side of commercial PP separator (Celgard 2400). The CoFe@FeO_x/PP (CoFe@C/PP, MnFe@FeO_x/PP or NiFe@FeO_x/PP) separator was obtained after vacuum-dried at 70 °C for all night, followed by cutting into disks with 16 mm in diameter. Similarly, the CoFe@FeO_x/GF (CoFe@C/GF, MnFe@FeO_x/GF or NiFe@FeO_x/GF) separator was prepared with the same process except that replacing PP separator with commercial glass fiber separator (Whatman GF/A 1823-070).” (Page 17, line 1)

Comment 4:

How does CoFe@FeO_x contribute to minimizing the polysulfide shuttling effect? What would happen with only CoFe/PP or FeO_x/PP?

Response 4:

Thanks for the reviewer’s valuable comments. The core@shell CoFe@FeO_x nanoparticles with CoFe alloy core and FeO_x shell exhibit bifunctional effect on regulating polysulfides. In detail, the polar FeO_x shell could effectively adsorb polysulfides in the surface, and the conductive CoFe core facilitates the conversion process of polysulfides, thus significantly suppressing the polysulfide shuttling effect. Both the shuttle current measurement (Figure S27) and Li₂S precipitate experiments

(Figure S28) demonstrate that the CoFe@FeO_x nanoparticles contribute to minimizing the polysulfide shuttling effect. It should be noted that the core@shell CoFe@FeO_x nanoparticle is prepared via one-step thermal syngas treatment, both single-component CoFe nanoparticle and FeO_x nanoparticle can not be obtained by this method. Because after the pyrolysis of Co-Fe Prussian blue analogue, the CoFe@C with a CoFe alloy core and a carbon shell are obtained rather than the single-component CoFe nanoparticle. And the outer thin FeO_x layer is formed through the passivation of Fe_xC shell in air. As a contrast, the cycling performance comparison of Li-S batteries with CoFe@C/PP separator and CoFe@FeO_x/PP separator are tested as exhibited in Figure R2 and Figure S24A. The Li-S battery with CoFe@FeO_x/PP can maintain a high reversible capacity of 913 mAh g⁻¹ after 100 cycles, however, the Li-S battery with the CoFe@C/PP keeps a reversible capacity of 631 mAh g⁻¹ after 100 cycles, which is poorer than that of the CoFe@FeO_x based battery.

Figure R2 Cycling performance comparison of Li-S batteries with different separators with 70 wt% of sulfur loading in the cathode at 0.2 A g⁻¹.

Following the reviewer’s comments, we made revision as below and in the revised Supporting Information:

“Detailed performance comparisons of Li-S batteries with traditional PP separator, CoFe@C/PP and CoFe@FeO_x/PP are exhibited in Figure S24A. The Li-S battery with CoFe@FeO_x/PP can maintain a high reversible capacity of 913 mAh g⁻¹ after 100 cycles. As a contrast, a low initial capacity of 595 mAh g⁻¹ is obtained with the traditional PP separator, demonstrating the CoFe@FeO_x can effectively anchor LIPSs and facilitate the conversion of sulfur species. And the Li-S battery with the CoFe@C/PP exhibits an initial reversible capacity of 993 mAh g⁻¹ at 0.2 A g⁻¹ and maintain a reversible capacity of 631 mAh g⁻¹ after 100 cycles, which is poorer than that of the CoFe@FeO_x based battery.” (Page 13, line 1)

Figure S24 Cycling performance comparison of Li-S batteries with different separators with 70 wt% of sulfur loading in the cathode at 0.2 A g⁻¹ (A). Rate capability comparison of CoFe@FeO_x/PP and CoFe@C/PP based Li-S batteries with 70 wt% of sulfur loading in the cathode at various current densities (B). Long-term cycling life of CoFe@FeO_x/PP based Li-S batteries with 70 wt% of sulfur loading in the cathode at 1 A g⁻¹ (C).

Comment 5:

How were the Li₂S precipitation experiments carried out?

Response 5:

Thanks for the reviewer's valuable comments. We have provided the Li₂S Precipitation Experiment in the Methods part.

Following the reviewer's comments, we made revision as below and in the revised manuscript:

"Li₂S Precipitation Experiment

The Li₂S precipitation experiment was tested on the 2032-type coin cells assembled with the tested electrode, lithium foil and PP separator on the Autolab PGSTAT 302N workstation. Li₂S₈ electrolyte (0.2 mol L⁻¹) was prepared by mixing sulfur with Li₂S at a molar ratio of 7: 1 in tetraglyme followed by vigorous mixing for all night. Commercial carbon papers (Guangdong Canrd New Energy Technology Co. Ltd.) were used as current collectors to load the well-mixed slurry composed of 70 wt% CoFe@FeO_x (or CoFe@C), 20 wt% Super P and 10 wt% CMC. The tested electrodes (loading mass: ~1.5 mg cm⁻²) were obtained after drying at 80 ° C for 12 h. 30 mL Li₂S₈ was dropped onto the tested electrodes during the cell assembly process. Cells were first discharged galvanostatically at 0.112 mA to 2.09 V and then discharged potentiostatically at 2.05 V for Li₂S nucleation and growth. The potentiostatic discharge was terminated when the current was below 10⁻⁵ A. Based on Faraday's law, the energy was collected to evaluate the nucleation/growth rate of Li₂S on the tested electrodes." (Page 17, line 10)

Comment 6:

The y-axis units in Figure S22 A, B, and Figure S26 appear incorrect.

Response 6:

Thanks for the reviewer's valuable comments. We have made a correction for original Figure S22 and Figure S26.

Following the reviewer's comments, we made revision as below and in the revised manuscript:

Figure S26 CV curves of CoFe@FeO_x/PP (A) and CoFe@C/PP (B) based Li-S batteries at various scan rates. The linear fitting plots of CoFe@FeO_x/PP and CoFe@C/PP based batteries (C, D).

Figure S31 The typical CV curves of RT Na-S battery with CoFe@FeO_x/GF at 0.2 mV s⁻¹.

Comment 7:

There is confusion regarding the sulfur loading. Somewhere authors claim 70 %, somewhere, 50 %. Authors should clearly mention the sulfur loading in every electrochemical measurement.

Response 7:

Thanks for the reviewer's valuable comments. We have clearly mentioned the sulfur loading in every electrochemical measurement in the revised manuscript.

Comment 8:

Authors claim shell is composed of graphitized and amorphous carbon. How have the authors confirmed that? What is the role of graphitized and amorphous carbon?

Response 8:

The CoFe@C presents core@shell structure, with a CoFe alloy core and a carbon shell, as shown in Figure R3. Also, we have added Raman spectra of CoFe@C and CoFe@FeO_x to the supporting information (Figure S2). The Raman spectra of the two samples exhibit two distinct peaks, in which the D-band (~ 1350 cm⁻¹) is ascribed to carbon with disorder/defects, while the G-band (~ 1590 cm⁻¹) reflects the carbon graphitization degree. Hence, the shell of CoFe@C is composed of graphitized and amorphous carbon according to the I_D/I_G value. The carbon shell may prohibit the aggregation of CoFe alloy core during pyrolysis process.

Figure R3 The TEM image of the CoFe@C with the size distribution histogram (A). HRTEM image and the corresponding lattice fringes (B) of the CoFe@C.

Following the reviewer's comments, we modified the discussion as below and in the revised manuscript:

“The Raman spectra of CoFe@C exhibits two distinct peaks with I_D/I_G of 1.00, indicating the shell presents a mixture of amorphous and graphitized carbon (Figure S2).” (Page 3, line 10)

Figure S2 Raman spectra of CoFe@C and CoFe@FeO_x.

Comment 9:

Authors have synthesized MnFe@FeO_x and NiFe@FeO_x; however, their performance is not shown for Li S or Na S batteries. Why so? What advantage do CoFe@FeO_x has over MnFe@FeO_x or NiFe@FeO_x

Response 9:

Thanks for the reviewer’s valuable comments. In this work, we developed a universal solid synthesis strategy to synthesize uniformly distributed core-shell MFe@FeO_x (M=Co, Mn, Ni) nanoparticles via thermal syngas treatment. As a case, we systematically studied the synthesis process of CoFe@FeO_x nanoparticle and application of CoFe@FeO_x in Li-S and RT Na-S batteries. And we also investigated the applications of MnFe@FeO_x and NiFe@FeO_x in Li-S and RT Na-S batteries in the revised manuscript.

Following the reviewer’s comments, we made revision as below and in the revised manuscript:

“Preparation of MFe@FeO_x (M=Co, Mn, Ni) modified separators

The CoFe@FeO_x/PP (CoFe@C/PP, MnFe@FeO_x/PP or NiFe@FeO_x/PP) separator was prepared by mixing 95 wt% of CoFe@FeO_x (CoFe@C, MnFe@FeO_x or NiFe@FeO_x) with 5 wt% Sodium Carboxymethyl Cellulose (CMC) binder in deionized water to cast on one side of commercial PP separator (Celgard 2400). The CoFe@FeO_x/PP (CoFe@C/PP, MnFe@FeO_x/PP or NiFe@FeO_x/PP) separator was obtained after vacuum-dried at 70 °C for all night, followed by cutting into disks with 16 mm in diameter. Similarly, the CoFe@FeO_x/GF (CoFe@C/GF, MnFe@FeO_x/GF or NiFe@FeO_x/GF) separator was prepared with the same process except that replacing PP separator with commercial glass fiber separator (Whatman GF/A 1823-070).” (Page 17, line 1)

“In addition, both the MnFe@FeO_x and NiFe@FeO_x based Li-S batteries exhibit good cycling stability as show in Figure S29, suggesting MnFe@FeO_x and NiFe@FeO_x could also effectively restrain the shuttle of LiPSs.” (Page 6, line 18)

“Moreover, the MnFe@FeO_x/GF and NiFe@FeO_x/GF based RT Na-S batteries could also deliver high capacities of 717 and 666 mAh g⁻¹ after 70 cycles at 0.2A g⁻¹, respectively, indicating both the

MnFe@FeO_x and NiFe@FeO_x can also suppress the shuttle effect of NaPSs and improve the utilization of S.” (Page 7, line 7)

Figure S29 Cycling performance of MnFe@FeO_x/PP (A) and NiFe@FeO_x/PP (B) based Li-S batteries at 0.2 A g⁻¹ with 70 wt% of S loading in the cathode.

Figure S36 Cycling performance of MnFe@FeO_x/GF (A) and NiFe@FeO_x/GF (B) based RT Na-S batteries at 0.2 A g⁻¹ with 50 wt% of S loading in the cathode.

Reviewer #2

This manuscript developed a solid synthesis strategy to precisely synthesize CoFe@FeO_x core@shell NPs. The introduction elaborated in detail the development of the traditional metal NPs synthesis strategies and the advantages of the solid synthesis approaches. The synthesis process of CoFe@FeO_x are presented carefully together with the structure and composition are characterized clearly. The synthesis mechanism is investigated according to the experimental results. This solid synthesis strategy is universal for the preparation of MFe@FeO_x (M=Co, Mn, Ni). The synthesized CoFe@FeO_x demonstrate high performance as the modifying layer of commercial separators in Li/Na-S batteries. This manuscript provides a novel perspective to synthesize complex metal NPs under thermal gas atmosphere, different from liquid environment of the traditional wet chemical methods. This solid synthesis strategy may enrich the current synthetic method of complicated structured metal NPs. Overall this manuscript is well written and I recommend to accept it after addressing following comments:

Comment 1:

It seems that the theoretical formation mechanism of the CoFe@Fe_xC confirms the

experimental synthesis of the CoFe@FeO_x. It is more reasonable to start with the synthesis and characterization of the CoFe@FeO_x, and then follow with the theoretical formation mechanism. The Co₂C phases are commonly observed in the syngas treated CoMn materials (Nature 2016, 538, 84-88.) For the CoFe alloy, the cobalt tends to be remained as alloy rather than converted into carbides. This interesting phenomenon is better to be explained with the physicochemical properties of Co and Fe.

Response 1:

We agreed with the reviewer that it is reasonable to start with the experimental section. The section of “Theoretical formation mechanism of the CoFe@Fe_xC” has been changed to follow the experimental sections in the revised manuscript. All figure numbers have been modified accordingly.

Indeed, Co₂C phases are commonly observed on the thermally syngas-treated cobalt-based materials. Co₃O₄ phase can be converted into Co₂C phase though syngas treatment at 250 °C. (ACS Catal., 2022, 12, 14, 8544–8557.) The Na promoted CoMnO_x phase can be converted into a mixture of Co₂C, Co, MnCO₃, MnO, and Mn_xCo_yO₄ through first reduction by H₂ at 350 °C and then treatment by syngas at 240 °C. (Nat. Commun., 2019, 10, 167.) Therefore, Co₂C phase tends to be formed through thermal treatment of cobalt and cobalt oxides by syngas. For CoFe alloy, Fe atoms are easy to be exsolved from the CoFe alloy nanoparticle bulk to the surface under CO adsorption (Figure R4A), and then the exsolved Fe shell is converted into Fe_xC shell due to the carburization ability of Fe is higher than that of CoFe alloy (Figure R4B).

Figure R4 The exsolution of Fe from CoFe alloy based on segregation energy (A). CoFe@Fe_xC formed through the carburization of Fe shell by CO (B).

Following the reviewer’s comments, we made revision as below and in the revised manuscript:

“It is interesting to observe that cobalt of CoFe@FeO_x remains as alloy phase in the core rather than converts into cobalt carbides, since Co₂C phase is commonly observed in the syngas treated CoMn materials³³. This can be explained by the exsolution of Fe from CoFe alloy due to the different segregation energy of Co and Fe.” (Page 4, line 41)

Comment 2:

The universal synthesis strategy section, MnFe@FeO_x and NiFe@FeO_x are also well characterized. Beside cobalt, what is the reason for the selection of Mn and Ni? Are they share

some similar properties? What is the most important factor to affect the formation of MFe@FeO_x core@shell structure? Can this synthesis strategies be extended to more bimetallic iron-based alloys? The forward-looking prospect may benefit the understanding of this synthesis strategy.

Response 2:

For CoFe@FeO_x, the formation of intermediate Fe_xC shell is important for the final core@shell structure. Iron carbides are commonly observed on the syngas treated iron-based materials. Hence, MnFe and NiFe alloy are selected to investigate whether they present similar structure evolution compared with CoFe alloy. In addition, the preparation of Ni-Fe PBA and Mn-Fe PBA is similar with that of Co-Fe PBA, except replacing the cobalt nitrate by nickel and manganese nitrate, respectively. The exsolution of Fe from CoFe alloy and the subsequent carburization of Fe shell are considered as the most important factor affecting the formation of MFe@FeO_x core@shell structure. Except CoFe, MnFe, and NiFe alloy, this synthesis strategies are supposed to be extended to CuFe and ZnFe alloy.

Following the reviewer's comments, we made revision as below and in the revised manuscript:

“Furthermore, this strategy is supposed to be extended to the synthesis of more iron-based bimetallic NPs with excellent distribution, such as, CuFe, ZnFe, etc. The most important factor of this strategy is the exsolution of iron from iron-based alloy and the subsequent carburization.” (Page 4, line 29)

Comment 3:

From Figure 2D and 4F, the FeO_x shell thickness of CoFe@FeO_x and NiFe@FeO_x are 5 and 2 nm respectively. What is the reason for this difference? The synthesis and characterization section mainly focuses on the description of the morphology, the structure, and composition of CoFe@FeO_x. The scientific reasons of these superficial phenomenon should be emphasized.

Response 3:

Yes, the different shell thickness of CoFe@FeO_x and NiFe@FeO_x is an interesting experimental result. Near equiatomic CoFe based alloys are face-centred cubic (fcc) (γ) above 983 °C, and body-centred cubic (bcc) (α) below this temperature. It is generally accepted that α orders to a B2 structure (α_2) below 730 °C (Figure R5).¹ NiFe alloy presents fcc crystal structure with Ni content in the range of 35-90% ((Figure R6).² We suppose the various crystal structure may lead to different segregation energy between Fe and Co/Ni. The difference value of the segregation energy between Fe and Co is higher than that between Fe and Ni. Therefore, less Fe atoms are exsolved from NiFe alloy to the surface compared with CoFe alloy, leading to thinner Fe shell and further to thinner FeO_x shell of NiFe@FeO_x.

Figure R5 The binary phase diagram for CoFe alloy. [Reprinted from¹. Copyright (2005), with permission from Elsevier.]

Figure R6 The binary phase diagram for NiFe alloy. [Reprinted from². Copyright (1983), with permission from Elsevier.]

Reference:

1. Sourmail, T., *Comments on Character of transformations in Fe–Co system. Scripta Materialia, 2005, 52, 1347-1351.*
2. Ferchmin, A. R.; Kobe S., *Amorphous magnetism and metallic magnetic materials-digest: a survey of the literature with a complete bibliography. 1983, North-holland publishing Co., Amsterdam, New York, Oxford.*

Following the reviewer’s comments, we made revision as below and in the revised manuscript:

“The FeO_x shell thickness of CoFe@FeO_x and NiFe@FeO_x are 5 and 2 nm respectively. CoFe and NiFe alloy present body-centred cubic and face-centred cubic structure, respectively, and the segregation energy may be influenced by the crystal structure. The thinner shell thickness of NiFe@FeO_x may come from the less difference of the segregation energy between Fe and Ni than that between Fe and Co, leading to less Fe atoms exsolved from NiFe alloy to the surface than from

CoFe alloy. ” (Page 4, line 23)

Comment 4:

The synthesized CoFe@FeO_x demonstrates high performance as the modifying layer of commercial separators in Li/Na-S batteries. The performance data of CoFe@FeO_x is explained in detail. Compared with the separator materials in the current literatures, what is the advantages of the CoFe@FeO_x ? What is the inspiration for future research? Furthermore, what about the performance of NiFe@FeO_x and MnFe@FeO_x in Li/Na-S batteries? Maybe some rules or trends of these bimetallic iron-based materials can be summarized according to their performance in batteries.

Response 4:

Thanks for the reviewer’s valuable comments. We have added related discussion associated with the advantages of the CoFe@FeO_x in the revised manuscript. And optimizing the specific proportion of CoFe and FeO_x is expected to further improve the utilization of S, which may be the inspiration for future research. We also investigated the applications of MnFe@FeO_x and NiFe@FeO_x in Li-S and RT Na-S batteries and concluded that the core-shell MFe@FeO_x ($M=\text{Co, Mn, Ni}$) can effectively enhance the performances of Li/Na-S batteries because of their unique bifunctional effect on regulating polysulfides.

Following the reviewer’s comments, we made revision as below and in the revised manuscript:

“Various conducting polymers and covalent–organic frameworks with strong chemical adsorptivity, or carbon matrices like carbon nanotubes or graphene with high conductivity have been employed as the modifying layer of commercial separators, and improving the utilization of S to some extent. However, most of these reported modifying layers hardly simultaneously possess strong adsorbability and high conductivity. The as-prepared CoFe@FeO_x exhibits bifunctional effect on regulating polysulfides as the separator coating layer for Li/Na-S batteries. In detail, the polar FeO_x shell could effectively adsorb polysulfides in the surface, and the conductive CoFe core facilitates the conversion process of polysulfides, thus significantly suppressing the polysulfide shuttling effect.”

“In addition, both the MnFe@FeO_x and NiFe@FeO_x based Li-S batteries exhibit good cycling stability as show in Figure S25, suggesting MnFe@FeO_x and NiFe@FeO_x could also effectively restrain the shuttle of LiPSs.” (Page 5, line 37)

“Moreover, the $\text{MnFe@FeO}_x/\text{GF}$ and $\text{NiFe@FeO}_x/\text{GF}$ based RT Na-S batteries could also deliver high capacities of 717 and 666 mAh g^{-1} after 70 cycles at 0.2A g^{-1} , respectively, indicating both the MnFe@FeO_x and NiFe@FeO_x can also suppress the shuttle effect of NaPSs and improve the utilization of S.” (Page 7, line 7)

“Thus, it can be concluded that the core-shell MFe@FeO_x ($M=\text{Co, Mn, Ni}$) can effectively enhance the performances of Li/Na-S batteries because of their unique bifunctional effect on regulating polysulfides.” (Page 7, line 11)

Figure S29 Cycling performance of MnFe@FeO_x/PP (A) and NiFe@FeO_x/PP (B) based Li-S batteries at 0.2 A g⁻¹ with 70 wt% of S loading in the cathode.

Figure S36 Cycling performance of MnFe@FeO_x/GF (A) and NiFe@FeO_x/GF (B) based RT Na-S batteries at 0.2 A g⁻¹ with 50 wt% of S loading in the cathode.

Comment 5:

Some minor errors should be corrected. Page 2, line 25, a space should be added between CoFe@FeO_x and core@shell. Page 3, line 33, Prussian blue analog is better to be replaced by Prussian blue analogue. Page 5, line 37, 0.2 mV/s should be revised as 0.2 mV s⁻¹. The text font in Figure 5 is too small. The manuscript should be fully checked carefully.

Response 5:

All errors and typo have been corrected accordingly in the revised manuscript. The full text has been carefully checked and proof reading by the native speakers.

Reviewer #3

Chen et al. have developed a universal thermal syngas treatment method for synthesizing MFe@FeO_x (M=Co, Ni, Mn). The obtained CoFe@FeO_x nanoparticles have been thoroughly characterized. Furthermore, these CoFe@FeO_x nanoparticles have been utilized as a separator coating layer in Li/Na-S batteries. This research work is highly intriguing and well-organized. The reviewer recommends publication of this work after making minor revisions.

Comment 1:

The article proposed the synthesis mechanism of CoFe@FeO_x according to the theoretical calculation. The segregation pathway of cobalt atom included four steps and that of iron atom is five steps. The segregation occurs with the exchanges between the cobalt/iron atom and surface/subsurface vacancies. I wonder where the initial vacancies come from. The theoretical calculation is supposed to verify the experiment results. For the experimentally prepared CoFe alloy NPs, do the vacancies appear in the CoFe crystal structure? I consider, whether this theoretical model can truly reflect the transformation of the CoFe crystal.

Response 1:

Thanks for the valuable comments, the corresponding contents have been added in the Supporting Information. Meanwhile, these new references (*Mater. Trans.* **2006**, *47*, 2646–2650; *Sci. Rep. UK* **2018**, *8*, 9764; *Int. J. Mater. Res.* **2022**, *97*, 861–871; *Phys. Rev. B* **2006**, *74*, 174108; *Phys. Rev. B* **2001**, *64*, 132102) have been cited and renumbered as the Refs. [1-5], and the original references [1-3] are renumbered as the Refs. [6-8].

Following the reviewer's comments, we made revision as below and in the revised manuscript:

“In fact, previous studies have confirmed that B2 CoFe alloy prepared by the mechanical alloying showed the presence of vacancy, antisite and point defects.¹⁻⁵ For example, Mizuno et al.¹ found that B2 CoFe alloy existed in a wide range of Fe composition from 30 to 75 at% at 773 K, however, aiming at compensating the deviation from the stoichiometric composition, constitutional defects are introduced; the formation energies of the vacancy and antisite defect in the CoFe alloy with different Fe compositions showed that both the vacancy and antisite defect are prone to be formed in the CoFe alloy with 50 at% Fe. Meanwhile, the chemically synthesized B2 CoFe nanoparticle also has defect owing to its synthesis method of one pot polyol process using ethylene glycol as a reducing agent, resulting in the disordered nature.² Moreover, the vacancies are also experimentally observed in the FeCo alloy, and the self-diffusion of the metals in both the disordered (A2) and ordered (B2) phase CoFe alloy occurs through the vacancies.³ Furthermore, Fu et al.⁴ theoretically studied the structural stability, point defects and order-disorder transition of B2 CoFe alloy, suggesting that B2 CoFe alloy is marginally stable, weakly ordered with a high density of antisite defects. Neumayer et al.⁵ concluded the presence of vacancies in the CoFe alloy based on ab initio statistical mechanics. Above these previously reported studies showed the presence of vacancies in the CoFe alloy, as a result, in our present study, B2 CoFe alloy is employed to explore its Fe segregation, in which B2 CoFe alloy with the vacancy is considered.

On the other hand, the vacancies in the CoFe alloy cannot be well characterized experimentally in our studies, however, our experiment results showed that Fe atoms would aggregate on the CoFe alloy surface, which means that Fe atoms easily segregates from the bulk to the surface in the CoFe alloy, however, in order to take place the segregation of Fe atoms from the bulk to the surface, only the presence of vacancies in the CoFe alloy could initiate Fe segregation, and realize the segregation of Fe atoms from the bulk to the surface. Moreover, theoretical calculation models of CoFe alloy also further verified that only the presence of vacancies in the CoFe alloy could realize the occurrence of Co/Fe segregation pathway from the bulk to the surface.

Furthermore, the alloy segregation takes place the exchanges between metal ions and surface/subsurface vacancies.⁶⁻⁸ For example, Kim et al.⁶ theoretically found that Au vacancy greatly accelerated the exchanges between Pd and Au in the PdAu alloy, meanwhile, surface Pd segregation induced by CO adsorption on the PdAu alloy surface would become more prominent. DFT studies by An et al.⁷ investigated Pd surface segregation in the AuPd alloy with the presence of CO, suggesting that Au vacancy is beneficial for promoting the exchanges between Pd and Au, leading to Pd surface segregation. Moreover, Zhang et al.⁸ found that NiAu core-shell structure exhibited a highly selective CO production in CO₂ hydrogenation due to the formation of a transient reconstructed NiAu alloy surface, in which Ni atoms offer active sites for CO₂ hydrogenation and the surface Au atoms contribute to the selective production of CO; meanwhile, aiming at analyzing the reconstruction of NiAu alloy surface, DFT calculations are adopted to consider the segregation pathway of Ni in the NiAu alloy, then, the model of NiAu alloy is constructed, in which an Au vacancy was set initially at the second atomic layer owing to the easy formation of Au vacancy, and the possible segregation pathway for a third-layer Ni atom near the vacancy was investigated to change the position of Ni through a series of the exchange steps between Au/Ni atom and the vacancy exchange steps.

Based on above analysis, the presence of vacancies in the CoFe alloy was confirmed, meanwhile, similar to above reported studies by Zhang et al.,⁸ in our present study, a Fe or Co atom vacancy in the CoFe alloy was set initially at the second atomic layer, then, the possible pathway for a third-layer Fe or Co atom near the vacancy was proposed to change the position of Fe or Co atom through a series of the exchange steps between Fe/Co atom and the vacancy, which could realize Fe/Co atom segregation.” (SI, Page 10-11)

“Reference:

1. Mizuno, M.; Araki, H.; Shirai, Y., *First Principles Calculation of Defect and Magnetic Structures in FeCo*. *Mater. Trans.* **2006**, *47*, 2646–2650.
2. Rajesh, P.; Sellaiyan, S.; Uedono, A.; Arun, T.; Justin Joseyphus, R., *Positron annihilation studies on chemically synthesized FeCo alloy*. *Sci. Rep. UK* **2018**, *8*, 9764.
3. Seeger, A., *Ordering processes and atomic defects in FeCo*. *Int. J. Mater. Res.* **2022**, *97*, 861–871.
4. Fu, C. L.; Krčmar, M., *First-principles study of the structural, defect, and mechanical properties of B2 FeCo alloys*. *Phys. Rev. B* **2006**, *74*, 174108.
5. Neumayer, M.; Fähle, M., *Atomic defects in FeCo: Stabilization of the B2 structure by magnetism*. *Phys. Rev. B* **2001**, *64*, 132102.
6. Kim, H. Y.; Henkelman, G., *CO Adsorption-driven surface segregation of Pd on Au/Pd bimetallic surfaces: Role of defects and effect on CO oxidation*. *ACS Catal.* **2013**, *3*, 2541–2546.
7. An, H.; Ha, H.; Yoo, M.; Kim, H. Y., *Understanding the atomic-level process of CO-adsorption-driven surface segregation of Pd in (AuPd)₁₄₇ bimetallic nanoparticles*. *Nanoscale* **2017**, *9*, 12077–12086.
8. Zhang, X. B.; Han, S. B.; Zhu, B. E.; Zhang, G. H.; Li, X. Y.; Gao, Y.; Wu, Z. X.; Yang, B.; Liu, Y. F.; Baaziz, W.; Ersen, O.; Gu, M.; Miller, J. T.; Liu, W., *Reversible loss of core-shell structure for Ni–Au bimetallic nanoparticles during CO₂ hydrogenation*. *Nat. Catal.* **2020**, *3*, 411–417.”

(SI, Page 10, line 8)

Comment 2:

The preparation process of CoFe@FeO_x is presented detailedly in Figure 1A. The structure and phase composition of CoFe@C is exhibited in Figure 2C and Figure 4A, as well as that of CoFe@FeO_x is displayed in Figure 2D, Figure 2E, Figure 3, Figure 4B. However, the intermediate of $\text{CoFe@Fe}_x\text{C}$ show less characterization information. Is there some related experimental evidence about this intermediate.

Response 2:

The intermediate of $\text{CoFe@Fe}_x\text{C}$ can't be characterized by HRTEM, XRD, and Mössbauer, due to its instability in air. There are three experimental evidences about this intermediate. The first experimental evidence is the Mössbauer result of CoFe@FeO_x . 7 wt% of Fe_xC can be assigned to the residue Fe_xC after air passivation (Figure R1B). The second experimental evidence is the GC results of the exhaust during carburization. Iron carbides are considered as the active phases in Fischer-Tropsch synthesis (FTS). The carburization process is also a catalytic process of FTS. The exhaust contains methane, ethylene, ethane, propylene, propane, butene, butane, pentene, and pentane (Figure S14), indicating the occurrence of catalysis and further indicating the existence of active Fe_xC phase.

The third experimental evidence is the air passivation of Fe_xC . Iron carbides are not stable in air and tends to be passivated to form an outside iron oxide shell, as shown in Figure R7 and R8. Normally, the iron oxide shell formed by air passivation is usually amorphous, however, it could be transformed to crystalline magnetite (Fe_3O_4) under electron beam irradiation during TEM analysis. (Ind. Eng. Chem. Res., 1996, 35, 1747–1752.; Catal. Lett., 1996, 37, 101–106.; Appl. Catal. A, 1999, 186, 277–296.; J. Catal., 2000, 196, 8–17.) Therefore, the CoFe@FeO_x are formed from the carburization of $\text{CoFe@Fe}_x\text{C}$.

Figure R7 HRTEM of iron nanoparticles after syngas treatment at 450 °C and the subsequent air passivation. (Stud. Surf. Sci. Catal., 1996, 101, 1421.)

Figure R8 HRTEM of iron nanoparticles after syngas treatment at 240 °C and the subsequent air

passivation. (Appl. Catal. B-Environ., 2022, 312, 121393.)

Following the reviewer's comments, we modified the corresponding discussion as below and in the revised manuscript:

"The experimental evidence of this intermediate is the GC results of the exhaust during carburization. The exhaust contains methane, ethylene, ethane, propylene, propane, butene, butane, pentene, and pentane (Figure S14), indicating the occurrence of catalysis and further indicating the existence of exposed Fe_xC phase. Fe_xC are considered as the active phases in Fischer-Tropsch synthesis (FTS). The carburization process is also a catalytic process of FTS.²⁵" (Page 4, line 4)

Figure S14 The GC-FID results of the exhaust during carburization.

Comment 3:

It is better to make a brief introduction to the development status of the current separator materials for Li/Na-S batteries. Whether literatures reported before about the application of core@shell metal NPs in batteries. What are the unique advantages of the CoFe@FeO_x core@shell structure?

Response 3:

Thanks for the valuable comments, we have made a brief introduction to the development status of the current separator materials for Li/Na-S batteries in the revised manuscript. Some literatures have reported the application of core@shell metal NPs in alkali ion batteries¹⁻⁴, however, the application of core@shell metal NPs, especially the core@shell metal NPs prepared via thermal syngas treatment have seldom reported as the modifying layer of commercial separators for Li/Na-S batteries. The unique advantage of the CoFe@FeO_x core@shell structure lies in bifunctional effect on regulating polysulfides. In detail, the polar FeO_x shell could effectively adsorb polysulfides in the surface, and the conductive CoFe core facilitates the conversion process of polysulfides, thus significantly suppressing the polysulfide shuttling effect.

Reference:

1. Kim, Y. B.; Seo, H. Y.; Kim, S. H.; Kim, T. H.; Choi, J. H.; Cho, J. S.; Kang, Y. C.; Park, G. D., Controllable Synthesis of Carbon Yolk-Shell Microsphere and Application of Metal Compound-Carbon Yolk-Shell as Effective Anode Material for Alkali-Ion Batteries. *Small Methods* **2023**, *7*, 2201370.
2. Yu, L.; Hu, H.; Wu, H. B.; Lou, X. W., Complex hollow nanostructures: synthesis and energy-related applications. *Adv. Mater.* **2017**, *29*, 1604563.
3. Wu, C.; Tong, X.; Ai, Y.; Liu, D.-S.; Yu, P.; Wu, J.; Wang, Z. M., A review: enhanced anodes of li/Na-ion batteries based on yolk-shell structured nanomaterials. *Nano-Micro Lett.* **2018**, *10*,

1-18.

4. Wang, S.; Zou, X.; Li, C.; Zheng, H.; Li, B.; Zhuang, Q., *Novel synthesis and electrochemical properties of fluoride cathode with a yolk-shell structure for K-ion battery. J. Power Sources* **2021**, *495*, 229721.

Following the reviewer's comments, we made revision as below and in the revised manuscript:

“Various conducting polymers and covalent–organic frameworks with strong chemical adsorptivity, or carbon matrices like carbon nanotubes or graphene with high conductivity have been employed as the modifying layer of commercial separators, and improving the utilization of S to some extent. However, most of these reported modifying layers hardly simultaneously possess strong adsorbability and high conductivity. The as-prepared CoFe@FeO_x exhibits bifunctional effect on regulating polysulfides as the separator coating layer for Li/Na-S batteries. In detail, the polar FeO_x shell could effectively adsorb polysulfides in the surface, and the conductive CoFe core facilitates the conversion process of polysulfides, thus significantly suppressing the polysulfide shuttling effect.”
(Page 5, line 37)

Comment 4:

Cobalt atoms are yellow and iron atoms are pink in Figure 1A, while cobalt atoms are green together with iron atoms are red in Figure 1B. The cobalt and iron atom color should be consistent with each other. The Fe_xC shell of CoFe@Fe_xC scheme is blue in Figure 1D, while it is brown in Figure 2A. The FeO_x shell is blue in Figure 2A and 2D, while it is brown in Figure 5D. The color of Fe_xC and FeO_x shell in the scheme should be consistent with each other.

Response 4:

The color of cobalt and iron atoms in Figure 1A and Figure 1B is modified to be consistent with each other. The color of CoFe@Fe_xC and CoFe@FeO_x scheme is also revised to be consistent in the manuscript.

Reviewer #4

The authors reported a solid synthesis strategy route for the synthesis of complex metal NPs by using syngas as medium, providing a totally different synthetic route from the traditional liquid-phase medium. Such gas-phase medium route allow them to prepare CoFe@FeO_x under syngas at 240 °C, this temperature is remarkably lower than the traditional CVD method. The authors demonstrated the carbon atoms of the syngas could be uniquely introduced into the metal crystal to form metal carbides intermediates, which are used to regulate the final metal structure. This method brings new concepts for the structural design of metal NPs. While the concept of achieving precise NP structure design and uniform dispersion is intriguing, several issues can be raised regarding the presented work. Below please find the comments.

Comment 1:

The authors emphasize the comparison between the traditional wet chemistry methods and this thermal gas treatment method in the introduction. The current development of thermal gas treatment method for preparing materials is suggested to added.

Response 1:

Thank you very much for your suggestions. We have added the current development of thermal gas treatment method for preparing metal materials in the introduction.

Following the reviewer's comments, we have added the content as below and in the revised manuscript:

“Metal NPs with complex structure can be synthesized via solid synthesis with thermal gas treatment. The Fe₃C and Co₃C nanocrystallines confined in graphitic shells are synthesized through a chemical vapor deposition method, which is operated under a mixture of H₂, H₂O, and CH₄ at 850 °C.¹⁹” (Page 2, line 13)

Comment 2:

Whether the structure of CoFe@FeO_x is influenced by the syngas treatment temperatures. For example, when the treatment temperature increases to 300 °C, or even 500 °C, what will happen for the metal structure?

Response 2:

We have added HRTEM images of CoFe@FeO_x and the corresponding discussion with syngas treatment temperature at 300 °C and 500 °C in the revised manuscript.

Following the reviewer's comments, we made revision as below and in the revised manuscript:

“The CoFe@FeO_x core@shell structure remains unchanged with the carburization temperature increasing from 240 to 500 °C (Figure S6, S7).” (Page 3, line 21)

Figure S6 The TEM (A) and HRTEM (B) images of CoFe@FeO_x with syngas treatment temperature at 300 °C.

Figure S7 The TEM (A) and HRTEM (B) images of CoFe@FeO_x with syngas treatment temperature at 500 °C.

Comment 3:

Co and Fe are both excellent syngas conversion catalysts. The thermal syngas treatment can also be considered as a catalytic reaction process. Does the composition of the exhaust gas in Figure 2A be monitored during the preparation process? This may help to understand the CoFe structure evolution.

Response 3:

Yes, the thermal syngas treatment can also be considered as a catalytic reaction process. We have monitored the exhaust of carburization by GC. The GC-FID results exhibit the exhaust contains methane, ethylene, ethane, propylene, propane, butene, butane, pentene, and pentane (Figure S14), indicating the occurrence of catalysis and further indicating the existence of active Fe_xC phase.

Following the reviewer’s comments, we made revision as below and in the revised manuscript:

“The experimental evidence of this intermediate is the GC results of the exhaust during carburization. The exhaust contains methane, ethylene, ethane, propylene, propane, butene, butane, pentene, and pentane (Figure S14), indicating the occurrence of catalysis and further indicating the existence of exposed Fe_xC phase. Fe_xC are considered as the active phases in Fischer-Tropsch synthesis (FTS). The carburization process is also a catalytic process of FTS.²⁵” (Page 4, line 4)

Comment 4:

Syngas is a type of largely produced chemical raw gas. This thermal syngas treatment methods present facile operation, indicating potential large-scale production. The authors are suggested to make a reasonable perspective about this method.

Response 4:

The reasonable perspective is added in the revised manuscript.

Following the reviewer’s comments, we made revision as below and in the revised manuscript:

“The facile operation and the abundant syngas indicate this solid synthesis strategy is suitable for producing well designed iron-based NPs in large scale.” (Page 4, line 32)

Comment 5:

In introduction section, it likely lacks sufficient information regarding the specific limitations of wet chemistry methods in achieving both NP structure design and uniform dispersion simultaneously. It would be beneficial to elaborate on these limitations to provide a clearer context for the proposed solid synthesis strategy.

Response 5:

The discussion about limitations of wet chemistry methods in achieving both NP structure design and uniform dispersion simultaneously is added in the introduction section.

Following the reviewer’s comments, we made revision as below and in the revised manuscript:

“Although the wet chemical methods control the metal NPs structure well, yet their liquid-phase operation environment leads to wastage of solvent or water. The construction of complicated structure needs multi-step operation, such as the core@shell structure is realized through coating a shell outside the pre-prepared core³. It is hard to both achieve the structure design and uniform dispersion of metal NPs simultaneously^{6, 11, 18}.” (Page 2, line 9)

Comment 6:

The methodology proposed in this work raises concerns about the exsolution and carburization processes. The selective exsolution of Fe atoms from the CoFe alloy bulk to the surface and their subsequent carburization into a Fe_xC shell under thermal syngas atmosphere may benefit from a more detailed explanation. Additionally, the authors are suggested to provide information on the reaction conditions and detail process, which are crucial for reproducibility and assessing the feasibility of the proposed method.

Response 6:

Thank you for your suggestions. We have added the detailed discussion of the synthesis process in the revised manuscript.

Following the reviewer’s comments, we have modified the discussion as below and in the revised manuscript:

“The formed CoFe@C experience carbon shell falling off, Fe shell formation by exsolution, and Fe_xC shell formation by carburization, producing the CoFe@Fe_xC intermediate. The carburization occurs under syngas atmosphere at 240 °C. The final CoFe@FeO_x are obtained through the passivation of Fe_xC shell in air at RT.” (Page 2, line 43)

Comment 7:

It is essential or desirable to include experimental data or theoretical explanations that demonstrate the generalizability of the solid synthesis strategy for other metal compositions.

Response 7:

Thank you for your comments. The generalizability of the solid synthesis strategy for other metal compositions has been added.

Following the reviewer's comments, we made revision as below and in the revised manuscript:

"Furthermore, this strategy is supposed to be extended to the synthesis of more iron-based bimetallic NPs with excellent distribution, such as, CuFe, ZnFe, etc. The most important factor of this strategy is the exsolution of iron from iron-based alloy and the subsequent carburization." (Page 4, line 29)

Comment 8:

Additionally, the bifunctional effect of the CoFe@FeO_x NPs on regulating polysulfides as a separator coating layer for Li/Na-S batteries was studied, however, the authors did not well clarify why CoFe@FeO_x/GF can significantly enhance the specific capacity of Na-S batteries.

Response 8:

Thanks for the valuable comments. As displayed in Figure 5D, the polar FeO_x shell could effectively adsorb polysulfides in the surface, and the conductive CoFe core facilitates the conversion process of polysulfides, thus significantly suppressing the polysulfide shuttling effect and enhancing the sulfur utilization and electrochemical performance of RT Na-S batteries.

Following the reviewer's comments, we made revision as below and in the revised manuscript:

"The mechanism schematic of CoFe@FeO_x enhancing performance of RT Na-S batteries is illuminated in Figure 5D. The core-shell CoFe@FeO_x possesses bifunctional effect on regulating NaPSs, because the polar FeO_x shell could effectively anchoring polysulfides in the surface and the conductive CoFe core further catalyzes the transformation process of NaPSs, thus significantly enhancing the utilization of S and electrochemical performance of RT Na-S batteries." (Page 6, line 31)

REVIEWERS' COMMENTS

Reviewer #2 (Remarks to the Author):

All comments are well addressed. No further revision is needed. Recommend to accept as it is.

Reviewer #3 (Remarks to the Author):

The authors have addressed my questions, and now it can be published.

Reviewer #4 (Remarks to the Author):

The authors have done an excellent job revising the manuscript, and the revised version is improved and suitable for publication in Nature Communications.